# Synthetic Calcium–Phosphate Materials for Bone Grafting

**DOI:** 10.3390/polym15183822

**Published:** 2023-09-19

**Authors:** Oleg Mishchenko, Anna Yanovska, Oleksii Kosinov, Denys Maksymov, Roman Moskalenko, Arunas Ramanavicius, Maksym Pogorielov

**Affiliations:** 1Department of Surgical and Propaedeutic Dentistry, Zaporizhzhia State Medical and Pharmaceutical University, 26, Prosp. Mayakovskogo, 69035 Zaporizhzhia, Ukraine; dr.mischenko@icloud.com (O.M.); alexeykosinov10@gmail.com (O.K.); maximov.d@zsmu.edu.ua (D.M.); 2Theoretical and Applied Chemistry Department, Sumy State University, R-Korsakova Street, 40007 Sumy, Ukraine; 3Department of Pathology, Sumy State University, R-Korsakova Street, 40007 Sumy, Ukraine; r.moskalenko@med.sumdu.edu.ua; 4NanoTechnas-Center of Nanotechnology and Materials Science, Institute of Chemistry, Faculty of Chemistry and Geosciences, Vilnius University, Naugarduko Str. 24, LT-03225 Vilnius, Lithuania; 5Biomedical Research Centre, Sumy State University, R-Korsakova Street, 40007 Sumy, Ukraine; m.pogorielov@gmail.com; 6Institute of Atomic Physics and Spectroscopy, University of Latvia, Jelgavas Iela 3, LV-1004 Riga, Latvia

**Keywords:** regenerative dentistry, synthetic bone materials, composites, hydroxyapatite, biopolymers

## Abstract

Synthetic bone grafting materials play a significant role in various medical applications involving bone regeneration and repair. Their ability to mimic the properties of natural bone and promote the healing process has contributed to their growing relevance. While calcium–phosphates and their composites with various polymers and biopolymers are widely used in clinical and experimental research, the diverse range of available polymer-based materials poses challenges in selecting the most suitable grafts for successful bone repair. This review aims to address the fundamental issues of bone biology and regeneration while providing a clear perspective on the principles guiding the development of synthetic materials. In this study, we delve into the basic principles underlying the creation of synthetic bone composites and explore the mechanisms of formation for biologically important complexes and structures associated with the various constituent parts of these materials. Additionally, we offer comprehensive information on the application of biologically active substances to enhance the properties and bioactivity of synthetic bone grafting materials. By presenting these insights, our review enables a deeper understanding of the regeneration processes facilitated by the application of synthetic bone composites.

## 1. Introduction

Regenerative dentistry, a significant branch of regenerative medicine, focuses on various dental pathologies, including bone defects such as periodontitis, alveolar bone resorption, caries, and pulpal necrosis. These localized skeletal diseases have a direct impact on patients’ quality of life and healthcare resources. To comprehensively address these diseases, therapy should concentrate on both bone regeneration and tooth regeneration [1].

From a regenerative standpoint, the bone structure is a vascularized connective tissue that possesses an inherent ability to remodel in response to external and internal factors during skeletal growth and development, as well as regenerate after injuries and pathological conditions. These processes involve a series of complex intercellular and intracellular biological interactions among various cell types and molecular signaling pathways [2]. Bone fracture healing represents one of the most common forms of bone regeneration in clinical settings [3,4]. Significant bone loss often occurs in the craniofacial region due to factors such as tumors, traumatic injuries, periodontal disease, congenital anomalies, or resorption resulting from tooth loss. In the fields of oral and maxillofacial surgery and orthopedics, there are cases where bone regeneration is required in larger quantities, exceeding the normal self-repair capacity. For example, in the reconstruction of large skeletal defects, such as dental implants, or when the innate regenerative ability is impaired, as seen in osteoporosis, optimizing the regenerative process is crucial to enhancing the likelihood of treatment success [5]. Bone grafting for improving the healing of bone defects involves autografts or allografts [6,7]. However, it is important to consider that autograft treatment has limitations, including relatively high morbidity rates at the donor site and a shortage of available grafts. Allografts also face challenges related to vascularization and integration with the host bone [8]. On the other hand, bioengineered bone tissue has the potential to overcome these problems and disadvantages [9].

Bone formation occurs through the differentiation of osteogenesis precursor cells into either mesenchymal osteoblasts, which synthesize randomly intertwined bone, or superficial osteoblasts, which produce highly ordered lamellar bone [10]. Recent studies suggest that endochondral and intramembranous ossifications, traditionally associated with fetal bone development, occur postnatally—particularly during skeletal regeneration following injury [11]. During the healing process, the bone recapitulates fetal bone development to achieve complete tissue regeneration without the formation of a fibrous scar [12]. Several transitional tissue types are involved in this process, including fibrous callus, low-mineralized cartilage, and tissue bone, depending on the degree of mechanical action [13,14]. These intermediate tissue types provide initial mechanical stability and are eventually replaced by ordered lamellar bone [14]. While bone has an intrinsic ability to regenerate and heal [4] and undergoes constant remodeling [15], incomplete or irregular postnatal osteogenesis can still occur in the case of large bone defects. Impaired regeneration can be caused by significant trauma, tumor resections, skeletal abnormalities, infections, or systemic disorders such as osteoporosis [16].

Achieving controlled, managed, and complete bone formation is the long-desired goal of bone tissue engineering. Autologous bone grafts are commonly used in orthopedic and maxillofacial surgery due to their superior histocompatibility, structural support, and minimal risk of immunogenic response [17]. To gain a better understanding of biomaterials’ functionality, it is important to briefly consider the processes involved in the reparative phase of bone tissue regeneration. Hematoma formation occurs when blood cells accumulate at the site of a bone injury, preventing further bleeding. The constriction of blood vessels also helps suppress bleeding. Within a few hours after the fracture, a hematoma forms as a result of the accumulation of blood cells and plasma fibrinogen. The hematoma supplies the fracture site with various growth factors, initiating the subsequent regenerative processes. Although this hematoma-mediated stimulation is beneficial in the case of a fracture, under conditions of granulation tissue formation, the hematoma occupies the available space and hampers blood circulation, leading to a slowdown in the regeneration process. Eventually, the clot shrinks and undergoes proteolysis before the epithelium infiltrates it [18].

Treating large bone defects remains a significant medical challenge. Biomaterial implantation is considered an important approach to promoting bone repair, although its effectiveness still presents challenges [19,20,21,22]. Bone healing includes early inflammatory immune regulation, angiogenesis, osteogenic differentiation, and physicochemical and mechanical properties associated with bone formation [4,16,23,24,25,26,27,28]. However, the regulatory role of the immune system in the biomaterial-mediated microenvironment of bone is often overlooked, which can lead to undesirable bone repair effects [29]. Once implanted in the body, biomaterials interact with immune cells and can trigger an inflammatory response, with the types of cells involved and the duration of the response significantly impacting therapeutic outcomes, ranging from fibrosis formation to regeneration (osteogenesis and angiogenesis) [30]. An uncontrolled inflammatory response can disrupt bone homeostasis, resulting in delayed wound healing and bone regeneration. Conversely, a beneficial anti-inflammatory immune microenvironment modulated by biomaterials may enhance bone cell differentiation, improve blood vessel formation, and enable successful long-term implantation [31,32]. Skeletal tissue regeneration achieved through the combination of pre-grown cells and growth factors with an appropriate scaffold is a promising approach, but synthetic bone graft substitutes with inherent osteoinductive properties may offer a more comprehensive solution [33]. Currently, a wide range of biomaterials are used as matrices. Major concerns, such as increased operative time, limited accessibility, and the risk of donor site morbidity [34], have led to the development of synthetic bone graft substitutes [16]. 

Bone is composed of hydroxyapatite (HA) (69–80%), collagen (17–20 wt.%), and other substances (water, proteins, etc.) [35]. Composite materials based on biopolymers and calcium phosphates are widely used for bone replacement [36]. Natural polymers are a good substitute for synthetic polymers. Their composition mimics extracellular matrix (ECM); they are bioabsorbable, biocompatible, biodegradable, and able to adsorb bioactive molecules. Biopolymer composites for medical applications should have similarities to the complex architecture of the human body and polymer composites. Furthermore, biopolymer-based drug and bioactive agent release systems should be developed as multifunctional release systems for maintaining the functionality of biological molecules [37,38]. There are around 28 types of collagen, but the most prevalent type found in the ECM of tendon and bone tissues is type I collagen [39]. Hydroxyapatite is deposited into the holes of type I collagen in the process of bone biomineralization [40]. Therefore, due to their compositional and structural analogies to natural bones, the composites of collagen and HA are of special interest among all kinds of bone substitutes [38,41].

Natural bones are a complex assembly of parallel type I collagen nanofibrils and HA crystals precipitated on their surface [42]. Thus, composite materials for bone replacement should be included in the process of bone formation promoted by osteoblasts (mineralization) to create an environment for the crystallization of calcium phosphates [41]. Two types of cells are involved in the bone formation process: osteoblasts (bone-forming) and osteoclasts (bone-resorbing). During the process of ossification, osteoblasts secrete type I collagen with noncollagenous proteins such as osteocalcin, bone sialoprotein, and osteopontin. Osteoblast-secreted ECM may initially be amorphous and noncrystalline but gradually transform into more crystalline forms [43]. One of the main challenges to bone tissue engineering is to develop scaffolds with optimal mechanical properties, biodegradability, and appropriate architecture for cell colonization and organization, which can ensure the integration of a scaffold with host tissue [38,41]. It should be clarified that natural bone is a kind of nanocomposite material that is heterogeneous and anisotropic. Its main components have several structural levels, from macro to nanoscale. The structure from the outer dense/cortical bone to the inner spongi/trabeculae represents the levels of macro- and microstructure, respectively [44].

Nanocomposites, primarily composed of mineralized collagens and minerals, exhibit structural characteristics similar to those of bone at the nanolevel. Consequently, to translate these novel discoveries into practical clinical applications, it is highly recommended to replicate the natural functionality of bone using advanced technologies such as 3D bioprinting and electrospinning [45].

As a result, the definition of biomaterial has evolved from being a “non-viable material used in a medical device designed to interact with biological systems” [46] to a “material intended to assume a form that can actively guide, through interactions with living systems, the progression of any medical or diagnostic procedure” [47]. This expanded definition signifies the growth and advancement of biomaterial science and technology, emphasizing its multidisciplinary, interfunctional, and translational nature.

## 2. General Principles of Bone Substitute Synthesis

### 2.1. Inorganic Phases of Bone Substitute Materials: Calcium Phosphate Materials

Materials based on calcium phosphates are widely utilized for bone regeneration due to their similar chemical composition to the mineral component of bones. Clinical studies have confirmed that highly crystalline hydroxyapatite undergoes a slower transformation in bone tissue during the resorption process compared to highly dispersed materials, such as nanocrystalline calcium phosphates. A fundamental distinction between HA in bone tissues and chemically obtained HA is its ultra-dispersed (nano) structure, with nearly 25% of the atoms located on the surface of the crystallite.

These surface atoms play a direct role in the chemical and metabolic processes occurring on the surface, as well as the activation of osteosynthesis mechanisms. Consequently, a crucial factor in achieving an HA structure that closely resembles human bone tissue is the formation of nanostructured surface relief and nanoporosity on the substrate. This facilitates the deposition of nanosized crystals of HA, imparting unique physicochemical properties that significantly contribute to bioactivation mechanisms [48].

Calcium phosphate ceramics are naturally formed in the solid tissues of the human body through biomineralization. From a biocompatibility standpoint, artificial materials made from calcium phosphates should possess optimal chemical and physiological properties. However, biological calcium phosphates exhibit important characteristics such as poor crystallinity, a high degree of element substitution within the composition, and very small crystallites that are often in close contact with the emerging protein matrix [49].

These distinctive features are closely related to the exceptional functional properties of mineralized tissues. Exploring new synthesis routes and processes to obtain biomimetic ceramics and composites based on calcium phosphates seems promising for advancing the synthesis and performance of bioceramics. Furthermore, the discovery of excellent biocompatibility and bioactivity properties in materials composed of the SiO_2_-CaO-P_2_O_5_ system has expanded the design possibilities and functionality of bioceramic materials [50].

Calcium phosphate composites enhance bioactivity, mechanical strength, elasticity, hierarchical structure, and porosity [51,52,53]. However, achieving materials that closely resemble the composition, structure, crystallinity, morphology, and biological properties of natural tissues is a challenging task for researchers. Nanosized low-crystalline hydroxyapatite, produced through wet chemical precipitation, shares chemical composition and particle size similarities with biological apatite but lacks essential substitution ions such as Na^+^, K^+^, Mg^2+^, and Cl^−^ and exhibits high reactivity [54].

It is important to note that not all calcium phosphate compounds have biomedical applications, as indicated in Table 1. Many synthetic calcium phosphates do not naturally occur in biological systems. In skeletal structures, they are predominantly found in the form of poorly crystallized, calcium-deficient apatite.

Compounds with an ionic ratio of Ca/P < 1 are unsuitable for implantation due to their high solubility and acidity. Similarly, TTCP (tetracalcium phosphate) is not suitable for medical use because of its basic nature. However, with proper combination and incorporation of other phosphates and chemicals, even these compounds can be successfully utilized in medical applications.

The solubility of calcium phosphate (CaP) phases is primarily influenced by their chemical composition, crystal properties, and the presence of cationic or anionic substitutions within the apatite lattice [58,59,60]. When comparing their dissolution in an acetate buffer, the solubility order is as follows: bone > enamel > β-TCP > HA. β-TCP demonstrates faster dissolution than HA in physiological solutions. The solubility of CaP ceramics is also influenced by factors, such as porosity and particle size. Higher porosity increases the surface area in contact with fluid, leading to an accelerated dissolution rate [61]. Hydroxyapatite, or calcium-deficient hydroxyapatite, is the most commonly used calcium phosphate for bone tissue regeneration. Brushite and octacalcium phosphate are also present in the human body. They are found in dental calculus and contribute to pathological calcification, as well as serving as intermediate compounds during the deposition of more thermodynamically stable HA during bone tissue mineralization [60].

Calcium phosphate materials with brushite (CaHPO_4_∙2H_2_O) as their main component are utilized for their in vivo resorbability [62,63,64]. Brushite, or calcium dihydrogen phosphate, can exist as an intermediate phase during HA precipitation and bone tissue mineralization. It remains stable in an acidic environment (pH < 6). The general reaction sequence is as follows: ACP → brushite → OCP → HA at pH = 6.5 and room temperature. Under physiological conditions, brushite can transform into HA in aqueous solutions [65,66,67].

#### 2.1.1. Brushite (Dicalcium Phosphate Dihydrate, DCPD)

Brushite has a monoclinic crystal lattice (Figure 1a), and hydroxyapatite has a hexagonal lattice (Figure 1b).

Dicalcium phosphate dihydrate is obtained by adjusting pH in the range of 3–4 at room temperature. DCPD can be produced from calcium-containing phosphates in a slightly acidic environment. It is often used as a component of bone cement and toothpaste to promote bone and tooth mineralization due to its biocompatibility, biodegradability, and osteoconductivity. DCPD could be converted to calcium-deficient hydroxyapatite in vivo. Brushite-based biomaterials are rapidly resorbed in vivo, realizing good biocompatibility in the absence of inflammatory cells.

Hexagonal modification, Ca_10_(PO_4_)_6_(OH)_2_, is idealized (P6_3_/m). At this structure, OH– crystal lattice nodes are placed on the screw axes of node 6_3_ (hexagonal axis c). In the HA structure, OH– crystal lattice nodes are placed above and below the mirror plane. This shift (~0.35 Å) alternates layer by layer according to the vertical axis in the above and below directions, transforming the 63 axis into 21, and the mirror plane into the b-axial plane [70]. 

Natural HA is usually calcium-deficient apatite with an atomic ratio Ca/P < 1.67 [48,71]. The general formula of naturally occurring HA is the following:

(Ca, M)_10_(PO_4_, Y)_6_(OH, X)_2_—where M—metal cations that include Mg^2+^, Na^+^, K^+^, Sr^2+^, Ba^2+^, etc., Y—anions CO_3_^2−^, H_2_PO_4_^−^, HPO_4_^2−^, SO_4_^2−^, etc., X—F, Cl, CO_3_^2−^, etc. [59]. It is therefore impossible to give a precise chemical formula for the mineral bone [61]. 

Substitutions of Ca^2+^, PO_4_^3−^, and OH^−^ ions in HA with elements M, Y, and X have a significant impact on various properties of the material, including lattice parameters, crystallinity, crystal symmetry, thermal stability, morphology, solubility, and both chemical and biological behavior [71]. For instance, the substitution of OH groups in HA with F^−^ ions enhances the structural stability and corrosion resistance of the material in biological environments. This substitution also promotes the growth of larger crystals and improves crystallinity [72]. Additionally, the substitution of cations can influence the properties of apatite. For example, the substitution of Ca^2+^ with Mg^2+^ ions results in a decrease in crystallinity and an increase in the solubility of HA [71].

The most mobile elements of the HA crystal lattice (Ca_10_(PO_4_)_6_(OH)_2_ = 2Ca_5_(PO_4_)_3_OH) are the Ca^2+^ cation and OH^−^ anion. They can easily move into an internodal position and create Frenkel point defects: cation V^2−^_Ca_^2+^ and anion V^+^_OH_^−^ vacancies.
Ca_10_(PO_4_)_6_(OH)_2_ ↔ Ca_9_V^2−^_Ca_(PO_4_)_6_V^+^_OH_ OH + Ca^2+^ + OH^−^_i_;
and internodal position of cations Ca^2+^ and anions OH^−^. 

Then, cations Ca^2+^_i_ and anions OH^−^_i_ move into physiological solution: Ca^2+^_i_ + OH^−^_i_ ↔ CaOH_s_^+^ ↔ Ca^2+^_aq_ + OH^−^_aq_,
where s is the position of the ion on the surface, and aq is the hydrated state of ions. 

Activated by vacancies, HA interacts with diffusing the lattice hydrogen proton H_i_^+^; as a result, the HPO_4i_^2−^ anion moved into the internodal position: Ca_9_V^2−^_Ca_(PO_4_)_6_V^+^_OH_ OH + H_i_^+^ → Ca_9_½V^2−^_Ca_H(PO_4_)_6_V^+^_OH_OH ↔ Ca_9_V^2−^_Ca_ (PO_4_)_5_V^+^_OH_ OH + HPO_4i_^2−^

Anion HPO_4i_^2−^ diffuses to the crystal surface and then moves into the physiological solution:HPO_4_^2−^ ↔ HPO_4_^2−^_4s_ ↔ HPO_4_^2−^_aq_

If the solution has a pH < 6, a primary orthophosphate ion is obtained:HPO_4_^2−^_aq_ + H^+^ ↔ H_2_PO_4_^−^_aq_

To complete the dissolution of HA, the overall reaction equation is used:Ca_10_(PO_4_)_6_(OH)_2_ + 14 H^+^ ↔ 10Ca^2+^ + 6H_2_PO_4_^−^ + 2H_2_O

The driving force of chemical dissolution (HA resorption) is the neutralization reaction of two hydroxyl groups of HA by two protons with a heat release of nearly 40 kJ/mol [48].

#### 2.1.2. Calcium-Deficient Apatite (CDA)

Calcium-deficient apatite (CDA) can be easily prepared by dropwise titration of a saturated solution of Ca(OH)_2_ with H_3_PO_4_ [73]. Another synthesis method consisted of adding calcium salt to a phosphate salt in basic media pH 11 buffered with ammonia (NH_4_OH) [74]. CDA crystals are poorly crystallized and are of submicron dimensions. The precipitated powders have large surface areas, typically 25 to 100 m^2^/g. Upon heating at 800 °C to 1000 °C, a particular composition of CDA leads to a pure b-tricalcium phosphate [β-Ca_3_(PO_4_)_2_] phase.

CDAs of various compositions can be precipitated in aqueous solutions. Depending on its composition, the calcium-deficient apatite decomposes at around 800 °C to 1000 °C, forming β-TCP and HA. At high temperatures, CDA with a Ca/P ratio of 1.58 leads to a mixture of HA and b-TCP in a weight ratio of 60:40, a so-called biphasic calcium phosphate [61]. 

Substitution of OH groups by CO_3_^2−^ leads to the formation of carbonate apatites, which are widespread in bone tissue. The CO_3_^2−^ group can substitute OH groups (an A-type substitution), as well as the substitution of PO_4_^3−^-3a (B-type). In general, substitution in carbonate apatite is mainly B-type [75]. 

HA is widely used clinically in bone regeneration as implants or coatings for other implants due to its good biocompatibility, bioactivity, and osteoconductive properties. Therefore, it is widely used for dental surgery, the repair of bone defects, vertebral fusion operations, and maxillofacial repairs. Mg-substituted HA in different forms displayed advanced bioactivity and promoted osteogenesis.

#### 2.1.3. Octacalcium Phosphate (OCP)

Octacalcium phosphate (OCP) is considered a mineral precursor to carbonate-containing calcium-deficient HA, which is a prototype for apatite crystals in bones and teeth. OCP has good osteoinductivity and is widely used in bone repair, including the coating of metal grafts, CaP bone cement, and scaffolds. In composite materials, OCP/collagen composite scaffolds have osteoconductivity that is positively correlated with the dose of OCP, and OCP acts as an initial deposition site for bone; its conversion to HA plays a significant role in bone formation. Moreover, osteoblasts that can initiate bone formation were found on the surface of OCP particles, and osteoblasts were directly attached to OCP to form bone matrix. The structure of OCP is closely associated with HA because OCP is composed of apatite layers stacked alternately with hydrated layers and can be converted into Ca-deficient HA in neutral aqueous conditions [38,76]. 

#### 2.1.4. Amorphous Calcium Phosphate (ACP)

Amorphous calcium phosphate (ACP) is often encountered as a transient phase during the precipitation of calcium phosphates (CaPs) in aqueous solutions. ACP formation is favored by rapidly mixing highly concentrated calcium and phosphate solutions at a high pH and low temperature. ACP forms at the beginning of precipitation due to its lower surface energy compared to octacalcium phosphate or hydroxyapatite. Over time, ACP can crystallize into calcium-deficient apatite through processes of internal hydrolysis and dissolution-reprecipitation.

The conversion of ACP into calcium-deficient apatite can be delayed by the presence of inhibitors of crystal growth, such as magnesium, pyrophosphate, or carbonate [77]. The exact chemical arrangement of atoms in ACP preparations is still uncertain, as many analytical methods do not provide precise crystallographic information. X-ray patterns typically exhibit a broad halo; infrared spectra show featureless phosphate absorption bands; and electron microscopy reveals spherical particles with diameters ranging from 20 to 200 nm and diffraction rings [59,60,77]. 

ACP plays an important role in the biomineralization process as a precursor of HA formation. ACPs express osteoconductivity and biodegradability, leading to a variety of applications, including CaP bone cement, scaffolds, bone repair biomaterials, and dental implants [48]. Nano-sized clusters in the ACPs have large specific surface areas and pH-responsive degradation, which makes them ideal drug delivery carriers. 

#### 2.1.5. Tricalcium Phosphate (TCP)

Tricalcium phosphate (TCP) cannot be directly precipitated from aqueous solutions. β-TCP can only be prepared by heating calcium-deficient apatite above 800 °C or through solid-state reactions. At temperatures above 1125 °C, β-TCP transforms into the high-temperature phase known as α-TCP. Although both compounds have the same chemical composition, they differ in their crystal structures [61].

In recent years, ceramics based on α and β-TCP have gained widespread use, primarily due to the higher solubility of TCP in contact with body fluids [78]. β-TCP exhibits lower solubility in water compared to α-TCP, which is more reactive in aqueous systems. When α-TCP comes into contact with water or body fluids, it undergoes rapid hydrolysis and reprecipitation as CDA, making it a valuable component in many calcium phosphate cements [61]. 

Compared to HA, β-TCP has better biodegradability and resorption rates, which can increase the biocompatibility of the implants. β-TCP has a relatively lower resorption rate than α-TCP, and the nanoporous structure of β-TCP provides excellent biomineralization, cell adhesion, and osteoblast proliferation. 

#### 2.1.6. Stoichiometric HA

Stoichiometric HA is the second-most stable and least soluble CaP after fluoroapatite. The preparation of pure HA from aqueous solutions is difficult owing to numerous ionic substitutions and possible lacunae in the crystal lattice. Some authors have reported its precipitation by slowly adding phosphate solution to the calcium solution and refluxing at 100 °C for 1 h [60,79]. After filtration, the precipitate is washed, dried at 80 °C, and heated at 800 °C to 1000 °C to form pure HA Ca_10_(PO_4_)_6_(OH)_2_. HA powder or slurry can be mixed with polymer spacers and heated in the range of 1000 °C to 1300 °C to form macro-porous ceramics [58,80].

In medical practice, hydroxyapatite bioceramics are used in the forms of powder, granules, solid material, porous material, part of composite materials, or a coating on various types of substrates [48,71]. HA is considered to have good biocompatibility, bioactivity, and osteoconductivity but expresses low osteoinductivity [58]. Therefore, it is better to combine HA with other materials for improved osteoinductivity. It has been applied in orthopedics, laryngology, dentistry, traumatology, maxillofacial surgery, and ophthalmology.

#### 2.1.7. Fluorapatite (FA)

Fluorapatite (FA) is the least soluble phase among calcium phosphates. It crystallizes in the same crystallographic system as hydroxyapatite, with fluoride ions substituting hydroxyl ions in the apatite tunnels [59,60]. FA readily forms a solid phase, but obtaining pure FA through precipitation in aqueous solutions is relatively challenging. Even at high concentrations of fluoride ions, the formation of solid-state solutions stabilized by hydrogen bonds between fluoride and hydroxyl ions in the apatite tunnels leads to the formation of fluoridated hydroxyapatite (FHA).

FHA is mainly found in bone tissue, and FA is found in tooth enamel. The presence of fluoride in saliva and plasma is necessary for dental and skeletal development. The physiological importance of fluorine ions in stimulating the mineralization and crystallization of calcium phosphates in bone formation has been proven. The osteoblastic response in terms of adhesion, differentiation, proliferation, and mineralization processes is enhanced by importing fluorine into hydroxyapatite compared to pure HA [60]. The amount of fluoride ions released directly affects the cell attachment, proliferation, morphology, and differentiation of osteoblast cells. FA has better protein absorption and cell attachment than HA [48]. 

HA is the most stable phase under physiological conditions and exhibits the slowest solubility and resorption kinetics in the human body. Implants made of sintered, pure HA ceramics can remain in bone defects for many years after implantation, indicating their non-resorbable nature. On the other hand, β-tricalcium phosphate (β-TCP) is resorbable, and the amount of remaining implant decreases over time. Biphasic calcium phosphate ceramics, which consist of a mixture of HA and β-TCP, are often preferred as bone substitutes. The solubility of BCP ceramics depends on the weight ratio of HA and β-TCP, making it closer to either β-TCP or HA [78,81].

Solid-state synthesis is a method used to obtain highly crystalline HA (up to 30 µm) with the required stoichiometry. However, it requires long-term annealing at temperatures up to 1300 °C under pressures of 20–31 kPa [78,81]. Among the methods for obtaining HA from aqueous solutions (Table 2), the most commonly used are the “wet methods”, which can be categorized into precipitation from solutions with constant or variable compositions, hydrothermal synthesis, and hydrolysis of calcium phosphates [78,81,82].

Fine crystalline precipitates of hydroxyapatite can be obtained during precipitation from alkaline aqueous solutions [78,81]. Studies have shown that prior to the formation of HA, the precipitation of amorphous calcium phosphate is observed [83,84]. In the initial stage, the sediment obtained often does not correspond to the exact composition of HA. However, when the primary precipitate of calcium phosphate is kept under appropriate conditions, the calcium-to-phosphate (Ca/P) ratio increases, and crystallization of HA takes place.

The crystallization rate of the primary HA precipitate is influenced by various factors, including the concentration of initial salts, the order and speed of reactant addition, mixing conditions, pH, reaction temperature, precipitation time, ionic strength of the solution, and the presence of impurities [85,86,87]. These factors play a crucial role in controlling the crystallization process of HA and achieving the desired crystalline structure and composition.

There are certain difficulties in the preparation of synthetic HA crystals, which are related to the chemical similarity of the material to some ions, the complex nature of calcium phosphate systems, and the role of kinetic parameters that depend on the experimental conditions [88]. 

The substitution of cations and anions in HA with appropriate groups present in bone tissue (Mg^2+^, Na^+^, K^+^, F^−^, Cl^−^, CO_3_^2−^, SO_3_^2−^) is used to regulate the chemical behavior of HA.

To obtain carbonate apatite ceramics, the following synthesis could be used [89].
10CaCl_2_ + 6(NH_4_)_2_HPO_4_ + x(NH_4_)_2_CO_3_ + (8 − 2x)NH_4_OH → Ca_10_(PO_4_)_6_(CO_3_)_x_(OH)_2−2x_ + 20NH_4_Cl + 6H_2_O (x ≤ 1),

In the final stage of the synthesis, a mixed solution containing (NH_4_)_2_HPO_4_ and (NH_4_)_2_CO_3_ is added to create equal conditions for the interaction of phosphate and carbonate ions with calcium and hydroxyl ions under constant stirring [90]. The degree of crystallinity of HA obtained through precipitation from aqueous solutions increases with higher temperatures and longer sediment aging times [81,91]. However, decreasing the preparation temperature, synthesis time, and initial solution pH can lead to deviations from the stoichiometric ratio of Ca/P = 1.67 in HA [92]. Controlling multiple parameters simultaneously can be challenging, which may result in poor reproducibility of morphology, particle size, and Ca/P ratio [92].

Synthetic HA (Ca_10_(PO_4_)_6_(OH)_2_) exhibits good stability in the body, while tricalcium phosphates (α-TCP, β-TCP, Ca_3_(PO_4_)_2_) are more soluble. Biphasic calcium phosphate, a mixture of HA and β-TCP, demonstrates intermediate properties depending on the weight ratio of stable and degradable phases. The dissolution rate follows the order: α-TCP > β-TCP > BCP > HA [93]. Recent advancements in the preparation of calcium phosphates have enabled the production of substituted CaP ceramics, which not only exhibit different solubility and bioactivity but also elicit modified biological responses through the release of biologically active ions during dissolution [94]. CaP ceramics can elicit various biological effects in vivo, and while most are osteoconductive, only certain types are osteoinductive. According to Samavedi et al., the osteoinductive potential of CaPs decreases in the order of BCP > TCP > HA [38,93].

High-temperature phases commonly used in medical applications, such as HA and α- and β-TCP in mono- or multi-phase forms [95], are obtained through a two-stage process: (1) synthesis of precursors and (2) high-temperature processing of these precursors. The biomimetic approach, which involves the use of simulated body fluids (SBFs) as a medium for ongoing processes, enables the production of bone-like nanosized materials with composition and ionic inclusions similar to solid tissues. Various simulated body fluids (EBSS, HBSS, SBFc, and SBFr) have been developed to mimic the composition of human extracellular plasma [96]. These fluids are complex, multicomponent systems containing Na^+^, K^+^, Ca^2+^, Mg^2+^, Cl^−^, HCO^3−^, HPO_4_^2−^, and SO_4_^2−^ ions and are supersaturated with phosphate and carbonate salts. As a result, they exhibit instability and create conditions for the simultaneous occurrence of stable and metastable crystallization processes, accompanied by dissolution, ion exchange, and other related phenomena. To explore these processes, thermodynamic calculations are applied to predict possible crystallization events in these systems, which then guide experimental investigations.

Initially, the amorphous calcium phosphate obtained in these systems transforms into a low-crystalline bone-like carbonate apatite when present in SBFc, SBFr, and SBFcg, with the rate of transformation being dependent on the composition of the specific simulated body fluid [97,98]. The fastest transformation is observed in SBFcg, followed by SBFr. The phase transformations of amorphous products result in changes in the composition of both the solid and liquid phases during maturation, highlighting the strong influence of the simulated body fluid’s composition on the dissolution/crystallization processes. The presence of glycine in SBFcg or higher concentrations of HCO_3_^−^ ions in SBFr leads to increased complexation in solution, thereby enhancing the dissolution rate and creating supersaturation conditions relative to the thermodynamically more stable phase.

The research results confirm the hypothesis that HA can crystallize in two stages:

(1) The initial precipitation of a metastable product and (2) the recrystallization of the latter over time into a thermodynamically more stable salt. 

The evidence presented suggests that the process of hydroxyapatite precipitation is primarily governed by kinetic factors rather than thermodynamic considerations. In the precipitation of Mg- or Zn-modified precursors, it was observed that all Zn^2+^ ions and approximately half of the Mg^2+^ ions from the reaction solutions were incorporated into the precipitated amorphous calcium phosphate. The different chemical behaviors of Zn^2+^ and Mg^2+^ ions can be explained by the “softness-hardness” factor and the crystal field stabilization energy (CFSE) [99].

The processing of these precursors, including filtration, washing, and calcination, can induce changes in crystal structure and agglomeration. To address this, new processing technologies need to be developed to preserve the initially obtained nanocrystals. Various biocompatible organic additives, such as natural polysaccharides (guar gum and xanthan gum), amino acids (glycine, alanine, and valine), and polyols (glycerin), have been proven to influence the composition and morphology of the obtained precursors, as well as the granulometric composition of their high-temperature phases [100]. The choice of modifiers aims to inhibit the growth of primary nuclei and improve the morphological characteristics of the particles. A biomimetic approach and continuous precipitation method were employed, wherein organic modifiers were added to the mother liquor and glycine buffer. The type, concentration, and maturation time of the organic additives play a role in shaping the initially deposited particles and their specific surface area. Low concentrations (<7 g/L) or the absence of additives, along with short maturation times, result in spherical particles with specific surface areas ranging from 28 to 48 m^2^/g. On the other hand, high concentrations of additives (>142 g/L) or longer maturation times lead to a loss of the spherical shape and elongation of particles, resulting in significantly increased specific surface areas (106–242 m^2^/g). While the specific surface area of the powders decreases sharply to 2–4 m^2^/g with an increased temperature in all cases, the powders obtained in modified media exhibit fine, unagglomerated particles with well-formed grains. The particle size distribution of samples heated at 1000 °C reveals that the narrowest range of particle sizes (0.25–0.55 mm) was achieved in a medium modified with xanthan gum.

### 2.2. Guiding Principles

#### 2.2.1. Heat Treatment of Precursors and Preparation of Final Fine Ceramic CaP Powders

To obtain ion-modified amorphous calcium phosphates, precipitation is followed by a stepwise calcination process. The amorphous precursors are subjected to calcination at temperatures of 200, 400, 600, 800, and 1000 °C, each temperature being maintained for 3 h. During calcination, the amorphous precursors transform into two-phase mixtures of HA and β-TCP, or monophasic magnesium (Mg)- or zinc (Zn)-modified β-TCP, depending on the concentration of Mg^2+^ and Zn^2+^ ions introduced into the structure. Both Mg and Zn substitutions contribute to the conversion of amorphous calcium phosphate to β-TCP, with the effect being more significant in the case of Zn substitution.

Monophasic Mg-β-TCP and Zn-β-TCP are observed at 600 °C for samples with Me/(Me + Ca) ratios above 0.05, while biphasic (HA and β-TCP) calcium phosphates are observed at ratios lower than 0.05. Rietveld refinement analysis confirms that Mg^2+^ and Zn^2+^ ions replace Ca^2+^ ions in the Ca(5) Mg/Zn-β-TCP octahedral positions, leading to a decrease in the average Ca(5)-O distances and the a and c cell parameters. The narrowing of the crystal lattice is more pronounced in Zn-substituted samples due to differences in the preferred coordination polyhedral for Zn and Mg ions [101].

The particle morphology of the calcined samples is influenced by both the deposition process and the post-treatment process. The standard approach, which includes filtration, washing, drying, and calcination, results in a dense, pore-free mass. However, to prevent particle agglomeration and obtain fine-grained unsintered ceramic powders with a composition close to that of hard tissues, a method involving several steps has been developed. These steps include biomimetic precipitation of ion-modified calcium phosphate precursors in various electrolyte systems, gelation of the suspension using xanthan gum, lyophilization at 56 °C, low-temperature (300 °C) calcination of the modified suspension for 1 h, washing the calcined sample with water, secondary gelation of the washed sample, lyophilization at 56 °C, and step sintering up to 1000 °C [102].

Selected ceramic powders with Zn/(Ca + Mg + Zn) ratios of 0, 0.01, 0.03, and 0.13, as well as Mg/(Ca + Mg + Zn) ratios of 0, 0.02, 0.05, and 0.10, were subjected to in vitro and in vivo testing to evaluate their cytotoxicity and response in bone tissue [102,103]. The results demonstrated the biocompatibility, osseointegration, and non-toxicity of the tested materials. They did not induce inflammation and only elicited a mild foreign-body reaction. The ceramic powders were found to be biodegradable within physiological limits, as evidenced by the presence of biochemical bone markers. These findings confirm the materials’ potential for bone regeneration and reconstruction.

To manufacture calcium phosphate materials with micro-, meso-, and macroporosity suitable for bone implants, samples were developed using natural polymers of different compositions and origins. Plant polysaccharides such as xanthan gum and carrageenan, along with animal gelatin, were utilized due to their water absorption and gradual decomposition properties within the body. These polymers provide sustained porosity in bone implants, supporting the growth of organic cells. A technology was developed to produce properly molded composite materials, involving the following steps: preparation of hydrogels with specific compositions; homogenization of gel-powder composite mixtures; appropriate shaping techniques; lyophilization; and modification of gelatin using a 1% glutaraldehyde solution. The optimal composite material consisted of Zn-modified β-tricalcium phosphate (Zn/(Ca + Mg + Zn) = 0.13) powder/gelatin/xanthan gum/carrageenan/water with weight proportions of 73.89/0.12/0.12/2.46/1.23/22.17 (wt.%). Subsequent storage of the composite material in simulated body fluid (SBF) for one month resulted in the formation of a new phase and partial dissolution of the polymers [104,105].

#### 2.2.2. Ordered Mesoporous Silicon–Calcium–Phosphate Composites

Ordered mesoporous ceramics belong to a class of porous materials characterized by uniform mesopores ranging from 2 to 50 nm and containing ordered mesostructures. The definition of these materials is primarily based on their physical sorption characteristics. According to the classification provided by the International Union of Pure and Applied Chemistry, porous solid materials are categorized as microporous if their pore diameter is up to 2 nm; macroporous if the pore size exceeds 50 nm; and mesoporous, indicated by the prefix “meso”, when the pore size falls between 2 and 50 nm [106]. While mesopores can be found in aerogels and columnar clays, which exhibit disordered pore systems with a wide pore size distribution, our focus lies on ordered mesoporous ceramics, specifically amorphous silicate materials synthesized in the laboratory, such as SBA-15 (Santa Barbara Amorphous No. 15) [107]. This material, first reported by Stucky et al. in 1998, consists of amorphous silica that forms cylindrical mesopores arranged in a hexagonal structure. An analysis of its low-angle X-ray powder diffraction confirms the presence of three peaks, with d-spacing values of 9.8, 5.6, and 4.8 nm, which correspond to the (100), (110), and (200) reflections of a two-dimensional hexagonal mesostructure with a lattice constant “a” of 11.2 nm in the space group P6 mm [108]. The synthesis of SBA-15 involves the use of a nonionic surfactant, specifically a block copolymer of polyethylene oxide and propylene oxide. This method leads to the formation of materials with mesopore sizes ranging from 5 to 30 nm and thicker wall structures compared to its homologous counterpart, MCM-41, which is synthesized using a cationic surfactant known as alkylammonium. 

In addition, the combination of amorphous silica composition and textural properties makes this material a very good candidate for medical applications as a biomaterial. According to the US Food and Drug Administration (FDA), silica has been “generally recognized as safe” and is particularly suitable as a biomaterial due to its high biocompatibility, non-toxicity of degradation products, and controlled hydrolytic degradation in biological environments [109]. In particular, amorphous silica particles decompose over time into the non-toxic orthosilicic acid Si(OH)_4_ and are excreted in the urine [109].

It is noteworthy that the SBA-15 type is obtained in the form of microparticles with a wide size distribution in the range from 1 to 100 microns. Some studies have evaluated the cytotoxicity of SBA-15 particles depending on their concentration in the medium. In our group, we observed that SBA-15 microparticles did not affect the viability of mouse macrophages up to concentrations of 100 μg/mL in cell culture medium [110]. However, higher concentrations compromised the viability of cells activated by Toll-like receptors [110]. Toll-like receptors (TLRs) are involved in host defense and autoimmune and inflammatory diseases.

The local effect of SBA-15 silica material has also been studied close to brain tissue. A cylinder of compressed material was surgically implanted in the temporal lobe of adult male rats and did not cause necrosis or inflammation, and the surrounding biological material self-adapted to the contour of the inorganic material [108]. 

#### 2.2.3. Growth of Hydroxyapatite Nanoparticles in Ordered Mesoporous Silica

Professor Hench’s discovery of bioglass in the 1970’s opened doors to a new category of biocompatible silica-based ceramics. He demonstrated that certain glasses with a predominant composition of silica in the ternary system SiO_2_-CaO-P_2_O_5_ can bond with bone tissue. These glasses were termed bioactive glasses, and Professor Hench defined a bioactive material as one that can spontaneously bind to living tissue without forming a fibrous interface or foreign body reaction capsule. The data obtained on bioactive glasses suggest that the ionic dissolution products of these glasses influence the cell cycle of osteogenesis precursor cells and possess osteogenic and angiogenic properties [111]. In this context, a material with a large surface area and an interconnected network of pores enhances the rate of ion exchange and dissolution, resulting in the release of constituent ions into the solution. Sol-gel synthesized glasses in the ternary system SiO_2_-CaO-P_2_O_5_ have demonstrated significant bioactivity [112]. Additionally, inspired by the bioinspired morphogenesis of bone-like hydroxyapatite nanoparticles using organic templates, researchers propose utilizing mesoporous silica to grow calcium phosphate apatite nanoparticles. They aim to achieve a high surface area in SiO_2_-CaO-P_2_O_5_ bioceramics. The concept involves initially incorporating calcium ions into the silica matrix as nucleation sites for the anchoring and growth of calcium phosphate crystals.

The synthesis procedure involved two steps: The low pH step entailed preparing a calcium-doped silica matrix, achieved by modifying the methodology of the standard SBA-15 material. The growth of hydroxyapatite within mesoporous silica was accomplished by the alkaline pH of the 9th stage, including phosphate ions [108]. The increase in pH induces condensation of neighboring silanol groups, leading to the formation of new oxo bridges and consequently a significant reduction in silanol groups. We hypothesized that these siloxane cavities could retain metallic calcium ions, stabilizing them in a coordination favorable for a macrocyclic effect. Such coordination can provide sites for the nucleation of hydroxyapatite crystals, ultimately filling the pores of the silica template.

NanoBone, a widely used commercial product in clinical practice [113,114], is a proprietary bone graft substitute composed of bioidentical nanocrystalline hydroxyapatite embedded in an amorphous silica gel matrix. The material is produced through a sol-gel process at temperatures up to 700 °C without undergoing sintering, resulting in a porous structure ranging in size from nanometers to micrometers. During surgical procedures, when in contact with the patient’s blood, approximately 80% of the volume becomes filled with the patient’s own proteins and biological material, effectively coating the entire inner surface area (approximately 84 m^2^/g). As a result, the biomaterial is perceived by the body as almost indistinguishable from the endogenous tissue [115]. 

In this context, the SBA-15-Nano HA material developed by our team consists of 20 nm HA nanoparticles incorporated into a mesostructured silica framework with a surface area of 275 m^2^/g. This material exhibits excellent adsorption properties, which make it highly suitable not only for adsorbing autologous proteins but also for potential applications in localized drug delivery following surgical interventions.

#### 2.2.4. Requirements for Calcium Phosphate Cements (CPCs)

Osteoinductance refers to the active induction of de novo bone formation [116]. Osteoconductivity, on the other hand, is the property that facilitates the colonization and ingrowth of new bone cells on the material’s surface. The osteoconductive nature of a material is primarily influenced by its chemical and physical properties, which promote cell adhesion and growth [117,118]. Osteogenicity, meanwhile, is associated with the presence of bone-forming cells within the bone graft. However, autologous bone grafting presents a significant drawback—the need for an additional surgical procedure to harvest a donor bone. Typically, an autologous donor bone is obtained from the iliac crest, which is easily accessible and contains a relatively high amount of cortical and cancellous bone [119]. Harvesting an autologous donor bone from the iliac crest can result in complications, with minor complications occurring in approximately 10% of cases and major complications in 5.8% of cases. Minor complications may include superficial infections, superficial seromas, and small hematomas. Major complications can involve vascular damage, deep hematomas requiring surgical intervention, deep infections at the donor site, fractures of the iliac bones, and neurological injuries [120], which may cause gait disturbances, shape deviations, paresthetic meralgia (neuropraxia of the lateral femoral cutaneous nerve), and protrusion of the intestine through an abdominal wall defect. An alternative treatment approach involves the use of allogeneic bone, where processed human cadaveric bone is transplanted into the patient. However, the allogeneic bone may not always be accepted as a bone substitute due to concerns such as adverse graft-versus-host reactions, graft necrosis, delayed engraftment, and relatively high costs [119,120]. Furthermore, both autologous and allogeneic bone grafts have limited availability [121].

Synthetic bone graft materials are actively being explored as an alternative to autologous and allogeneic bone grafts. They eliminate the need for a second surgical procedure and minimize the potential complications at the donor site. However, synthetic bone grafts do not possess all the characteristics of natural autologous bone, making continuous research efforts crucial for improving and developing an ideal material for bone replacement.

The Diamond concept [122,123] proposes that four essential parameters are necessary for unhindered fracture healing: osteogenic cells, an osteoconductive scaffold, growth factors, and a stable mechanical environment. Subsequently, vascularization at the site of the defect was recognized as an important factor in the fracture healing process [124].

The ideal bone graft material should fulfill these criteria. In this regard, various synthetic graft materials have been evaluated as scaffolds for bone restoration, with bioceramics being particularly appealing. Bioceramics can be classified into bioinert types (e.g., aluminum oxide or zirconia) and bioactive/bioresorbable types. Calcium phosphate-based bioceramics, calcium sulfate-based bioceramics, and silica-based bioactive glasses are among the extensively studied bioactive/bioresorbable bioceramics. Khabraken et al. [125] outlined the characteristics of an ideal bioceramic material for bone tissue engineering as follows:Biodegradable to support bone remodeling;Macroporous structure to facilitate tissue ingrowth;Mechanically stable and easy to handle;Osteoconductive, guiding bone growth around and within the material;Suitable for use as a carrier of growth factors or cells.

Synthetic biomaterials for bone regenerative treatments are employed due to their biological effectiveness, which is characterized by biocompatibility, bioactivity, and osteoconductive properties [126]. CaP-based bone substitutes promote attachment, proliferation, migration, and phenotypic expression of bone cells, leading to the formation of new bone in direct contact with the biomaterial [116]. CaP-based bone graft materials are commonly available as granules, blocks, and, more recently, cement. Among them, calcium phosphate cements (CPCs) are particularly attractive for clinical use due to their injectability and moldability, enabling minimally invasive application and optimal filling of irregularly shaped bone defects [127,128].

Contrarily, when using granules or blocks for implantation, they are typically mixed with a liquid (e.g., blood), resulting in suboptimal contact between the bone and the implant. Furthermore, blocks cannot be placed using minimally invasive surgery, and their size needs to be adjusted to match the defect through cutting, shaping, or drilling.

In bone regenerative procedures involving calcium phosphate cements (CPCs), the complete degradation and replacement of the CPC with a new living bone is preferred. However, as mentioned earlier, the biodegradability of CPCs is relatively low. Ideally, the rate of CPC biodegradation should closely match the rate of new bone formation to ensure the gradual restoration of mechanical properties in the newly formed bone tissue.

In vivo degradation of CPCs can occur through two different mechanisms: (1) Passive degradation due to the dissolution of the ceramic matrix in the extracellular fluid, and (2) active degradation mediated by cellular activity, including osteoclasts, giant cells, and macrophages. The rate of passive degradation, which involves the dissolution of the matrix in the extracellular fluid, depends on the CPC properties, such as surface area, calcium-to-phosphate (Ca/P) ratio, crystallinity, solubility, pH, and bodily fluid perfusion [118,129]. Previous studies [130,131] have shown that the physical destruction of CPCs can lead to ion dissolution and particle fragmentation due to the loss of mechanical integrity. On the other hand, the active degradation of CPCs is primarily mediated by giant cells and osteoclasts, with macrophages also playing a role in phagocytosing fragmented particles [132,133,134]. Macrophages have been observed to colonize the surface of CPCs shortly after implantation and are suggested to have a critical role in biodegradation [135]. Additionally, biomaterial particles released from CPCs can interact with immune cells, triggering the release of inflammatory mediators [136]. When macrophages encounter calcium phosphate particles, they attach to them and become activated for phagocytosis [129]. While macrophages are crucial for phagocytosing small fragments and particles, osteoclasts are primarily responsible for the active biodegradation of CPCs. These cells locally reduce the pH near the biomaterial, leading to the in vivo degradation of CPCs [137].

The presence of pores in calcium phosphate cements contributes to their degradation and various beneficial effects. Pores facilitate fluid flow, including perfusion in the case of interconnected porosity, as well as migration and proliferation of osteoblasts into the CPCs. Pores also promote vascularization and improve the stability of the tissue-implant interface by providing more surface area for cell proliferation and tissue regeneration. Pores in CPCs can be categorized based on their size as micropores (internal pore width <1 µm), mesopores (internal pore width 1–100 µm), and macropores (internal pore width >100 µm) [35,138]. The formation of microporosity in CPCs is attributed to the solidification mechanism, where crystals grow into needle-like or lamellar structures, creating a microporous structure [139]. The microporosity in CPCs can reach up to 60%, which increases the surface area, facilitates fluid penetration, and promotes protein adsorption [140].

The size of micropores can be controlled by adjusting processing parameters such as the particle size of the powder phase and the calcium-to-phosphate (Ca/P) ratio. Studies have shown that decreasing particle size leads to smaller pore sizes, and at a low Ca/P ratio, the pore size decreases due to reduced space between particles in the mixture [141]. The sintering temperature of the powder phase also affects microporosity, with higher temperatures resulting in less microporosity and changes in crystal size [142,143]. On the other hand, mesopores and macropores refer to pore sizes exceeding 1 µm and 100 µm, respectively. However, introducing meso- and macroporosity in CPCs requires specific methods. These larger pores are essential for cell migration, proliferation of osteoblasts, and mesenchymal cells, as well as promoting bone ingrowth.

High porosity and large pore sizes are known to enhance bone ingrowth into CPCs. Studies comparing porous and non-porous hydroxyapatite have shown that osteogenesis occurs in porous CPCs but not in solid particles [144]. A pore size of around 100 µm is generally considered sufficient for bone regeneration, as smaller pores may lead to ingrowth of unmineralized bone or fibrous tissue while hindering blood vessel ingrowth [145]. However, there is evidence suggesting that pores larger than 300 µm can also promote osteogenesis in certain cases, while pores smaller than 100 µm have been shown to facilitate bone formation or ingrowth into synthetic materials [146,147,148,149,150].

Another important aspect of CPC porosity is the connectivity of pores. Connectivity refers to the extent to which the introduced pores in CPCs are interconnected. Highly interconnected pores offer advantages over “dead-end” pores as they provide efficient pathways for fluid flow, cell migration, and distribution within the CPCs. Interconnectivity also promotes the formation of blood vessels, which are crucial for the development and remodeling of new bone tissue [151,152,153].

In summary, the presence of pores in CPCs, particularly micropores and interconnected porosity, plays a vital role in fluid flow, cell migration, vascularization, and promoting bone ingrowth and regeneration. Researchers continue to explore the optimal pore sizes, porosity levels, and interconnectivity to develop CPCs that effectively support bone healing and restoration.

The porosity and interconnectivity of calcium phosphate cements can be assessed using various approaches, including image-based methods and physical methods. Image analysis techniques utilizing scanning electron microscopy (SEM) or microcomputed tomography (micro-CT) are commonly employed for porosity and pore size measurements. SEM images are analyzed using software applications to quantify porosity and determine pore size [154,155]. Micro-CT imaging allows for the transformation of 2D X-ray images into 3D models, enabling the extraction of quantitative morphological data [156,157]. Physical methods for porosity assessment include gravimetry and mercury porosimetry. Gravimetry involves calculating the total porosity by comparing the density of the material comprising the CPC with the apparent density of the CPC itself [158,159]. Mercury intrusion porosimetry is a technique where mercury is injected into CPC constructs under increasing pressure. This method provides information about open and closed porosity (volume of mercury penetration into the CPC) as well as pore size (based on the decreasing radius of pores that can be filled as pressure increases) [159,160,161].

To enhance bone regeneration and address the limited degradation of CPCs, it becomes necessary to introduce macroporosity into the material. Macroporosity can be achieved through various methods such as the use of blowing agents, rapid prototyping techniques, or injection of blowing agents. However, it is important to consider that while increasing macroporosity is crucial for bone regeneration, it can simultaneously lead to a decrease in mechanical properties and changes in the manipulation properties of the CPC. Therefore, a compromise must be reached in the design of macroporous CPCs to balance macroporosity for bone regeneration with other important material properties. This ensures that the resulting CPCs exhibit the desired degradation behavior, mechanical stability, and handling characteristics.

#### 2.2.5. Foam Concentrates for Increasing Porosity of Calcium Phosphate Cement

Foaming is a viable method to introduce macroporosity into calcium phosphate cement and can be achieved through the generation of gas during a chemical reaction. Researchers have developed various techniques to create macropores in CPCs using different chemical reactions. Almirall et al. developed a technique using hydrogen peroxide decomposition to introduce oxygen macropores into α-TCP cement paste. By controlling parameters such as the Ca/P ratio and hydrogen peroxide concentration, they achieved high porosity levels of up to 66% [162].

Real et al. utilized an acid reaction between NaH_2_PO_4_ and NaHCO_3_ to generate CO_2_ bubbles within the CPC. This method resulted in porosity of up to 50% [161]. Studies in goats demonstrated that macroporous CPCs exhibited significantly greater bone formation compared to control CPCs without macroporosity. After 10 weeks, the macroporous CPCs were nearly completely degraded, and new bone formation was observed, while the control CPCs remained intact [163]. Similar positive results have been observed in animal models such as rats [164] and rabbits [165,166].

Another method involves the introduction of CO_2_ bubbles into CPCs through an acid-base reaction between NaHCO_3_ and citric acid monohydrate (C_6_H_8_O_7_·H_2_O). This approach yielded macropores with a size of 100 μm and macroporosity of up to 21% [167]. This method has been successfully employed in several studies, including the creation of pre-vascularized CPCs through the co-culture of endothelial cells and osteoblasts, which have potential applications in bone regeneration [168,169]. These studies demonstrate the effectiveness of foaming techniques in creating macroporous CPCs, and the resulting macroporosity has shown promising outcomes in promoting bone formation and tissue regeneration in various animal models.

The introduction of surfactants is another approach to creating foamed calcium phosphate cements. Surfactants can be either natural, such as albumin or gelatin, or synthetic, such as polysorbates. Synthetic surfactants, in particular, have been extensively studied and have been shown to produce highly porous CPCs with porosity levels of around 70%. These foamed CPCs exhibit osteoconductive and osteoinductive properties [170,171,172,173,174].

Biocompatible materials can be categorized into metals, bioglass, ceramics, and composites, which serve as stable replacements for various clinical applications such as maxillofacial surgery, implantology, neurosurgical skull reconstruction, and orthopedic surgery [12,175,176,177]. They are commonly used in the form of scaffolds (e.g., cementless prosthetic fixation, screws, fixation plates), intraosseous augmentation (e.g., cementoplasty, allograft), guided bone regeneration membranes, and other materials [178]. Optimizing the porous structure of bone substitutes is crucial for effectively regulating cellular responses in tissue engineering. Small pores are beneficial for cell attachment, but they may limit cell viability, proliferation, and differentiation [179,180]. On the other hand, highly porous biomaterials with larger pore sizes allow for better oxygen diffusion, which can enhance cell viability [181]. Graded pore bioceramics, particularly in the 500–800 µm size range, have shown significant improvements in cell adhesion, increased cell viability, and upregulation of angiogenesis-related gene expression, aligning with findings reported in the literature [182].

Angiogenesis, the formation of new blood vessels from existing ones, is a crucial process for successful implant integration and tissue regeneration [183]. The architecture of the pores within a porous implant plays a significant role in vascularization by providing space for tissue ingrowth and blood vessel formation. Studies have shown that macropores larger than 400 µm can promote the development of larger diameter blood vessels and reduce fibrous tissue ingrowth [184]. Larger pores are advantageous for delivering oxygen and nutrients to cells within the implant, thereby facilitating blood vessel formation. During the early stages, pore size influences the number of blood vessels formed. However, in later stages, pore size does not significantly affect the number of blood vessels, but it does impact their diameter. Small peripheral pores prevent the infiltration of large vessels into the central large pores, as reported in previous studies [185]. Additionally, the interconnection of pore windows within the scaffold can act as a bottleneck for vascular invasion, which affects scaffold vascularization [186,187].

The presence of small pore windows allows for the invasion of small blood vessels, but this pore architecture can restrict the penetration of blood vessels into the central region of the scaffold, regardless of the size of the pores in that region. Therefore, a graduated radial pore structure with a gradual increase in pore size from the center to the periphery is desirable for promoting vascularization of the implant. Rapid vascularization of an implant is a critical factor for successful clinical outcomes. It improves the integration of the implant with host soft tissues and reduces the risk of complications. As fibrovascular tissue grows into the macropores of a porous implant, soft tissues can be mechanically secured to the implant, reducing migration and exposure of the implant [188]. Abundant blood vessels also provide immune support, promoting wound healing and reducing the risk of postoperative infections [189]. Long-term clinical results have shown that coralline hydroxyapatite implants with 80% porosity achieve very fast vascularization rates, followed by synthetic hydroxyapatite implants (50–65% porosity), and finally Medpor implants (~41% porosity) [190,191].

Bone defects caused by various factors can have a significant impact on a patient’s quality of life [192,193]. Synthetic bone materials possess important characteristics such as protein adhesion, in vivo degradation, and osteoconductivity [194]. The induction of osteoinductivity can be achieved by creating a macroporous three-dimensional environment [195]. Several bioceramics are commonly employed in bone defect healing and bone tissue regeneration due to their excellent biocompatibility, osteoconductivity, and osteoinductivity. These include hydroxyapatite, β-tricalcium phosphate, akermanite, and 45S5 bioglass [196,197,198,199,200].

Three-dimensional (3D) porous scaffolds play a crucial role in bone defect repair and bone tissue engineering. They provide mechanical support, maintain tissue shape and integrity, promote bonding with surrounding tissues, and guide tissue growth [201,202,203]. The porous structure facilitates cell migration, growth, nutrient and metabolite transport, and stimulates bone integration and revascularization [202,204,205,206]. Additionally, some scaffolds can release biologically active ions that contribute to physiological stimulation of cells [201,204,205,206]. Therefore, 3D porous scaffolds are key elements in bone tissue engineering. Various technologies have been employed to manufacture bioceramic-based scaffolds based on specific characteristics and technical requirements. These include the template method, freeze-drying, foaming method, electroforming, and 3D printing [201,204,206,207].

#### 2.2.6. Calcium Phosphate (CaP) Ceramic Based Bone Grafts

Calcium phosphate ceramic-based bone grafts are gaining popularity in the field of bone grafting due to their chemical and biological similarities to the mineral phase of bone. Traditional CaP bioceramic therapy involves implanting bone grafts in the form of blocks or granules, which requires prior knowledge of the defect size and shape, followed by surgical implantation of appropriate bone substitutes [208,209].

To address the drawbacks of bone augmentation procedures, injectable bone cements have been developed and are receiving increasing attention due to their minimally invasive administration. Calcium phosphate composites are particularly known for their ability to self-align in vivo, making them advantageous for minimally invasive surgery [209,210,211]. In 1983, research by Brown and Chow led to the development of a new injectable form of CaP, which included tetracalcium phosphate, dicalcium phosphate dihydrate, CaHPO_4_·2H_2_O), and anhydrous dicalcium phosphate (CaHPO_4_) [212]. These developed materials exhibited properties such as self-hardening ability, good injectability, formability, increased reactivity, and high suitability for the development of new drug delivery systems [211,213].

To improve the cohesiveness and injectability of calcium phosphate ceramics, researchers have investigated the combination of CPC with polymer solutions and various additives [214,215,216]. Both natural and synthetic polymers have been incorporated as a liquid phase into injectable CPC to enhance adhesion, injectability, set time, and mechanical properties [215,217,218,219,220].

Chitosan, a natural amino-polysaccharide, has been used as a liquid phase additive to modify the physical properties of CPC, such as injectability, set time, and rheology while enhancing in vivo bioactivity [221]. Sodium alginate, collagen, gelatin, hyaluronic acid, and cellulose derivatives such as hydroxypropylmethylcellulose, methylcellulose, and carboxymethylcellulose have also been utilized as liquid phases for the formation of CPC [213,222,223,224,225,226,227,228,229,230]. The combination with biopolymers allows for the regulation of cohesiveness, injectability, mechanical properties, and bioactivity of the resulting cement.

Incorporating poly(lactic glycolic acid) microparticles into CPC has shown in situ macropore formation and increased cement strength, which is beneficial for bone reconstruction [231]. The addition of citric acid as a thinner has been shown to improve the injectability of CPC [232]. Studies have reported that citric acid can also enhance CPC set time and compressive strength, depending on the concentration of the additive [217,218].

Other attempts to regulate the physicochemical and biological properties of CPC include the addition of glycerol, strontium carbonate, polyethylene glycol, foaming agents, and β-dicalcium silicate [233,234,235,236,237]. These combinations and additives contribute to improving the performance and functionality of CPC for bone tissue engineering and regeneration applications.

In recent years, there has been growing interest in combining calcium phosphate cements with bioactive glasses (BGs) to enhance their properties. Bioactive glasses, such as 45S5 Bioglass^®^, have been known since 1969 and are composed of silicon oxide (SiO_2_), calcium oxide (CaO), phosphorus oxide (P_2_O_5_), and sodium oxide (Na_2_O) [238,239]. BGs can bond chemically to bone, promoting bone growth. The composition of BGs can be varied to create different variants by adjusting the basic SiO_2_-CaO-P_2_O_5_-Na_2_O ratio [238].

When BGs come into contact with bone tissue, they release silica ions from their surface. These ions form a layer of silica gel, followed by the precipitation of amorphous calcium phosphate and the subsequent formation of a layer of hydroxyapatite. This HA layer activates cell migration and promotes new bone formation, facilitating the integration of the BG with the surrounding bone tissue [240]. The high density of silanol groups (Si-OH) on the silica layer creates a negatively charged surface that plays a crucial role in inducing HA nucleation [239].

The dissolution products of BGs have been found to stimulate gene expression in osteoblastic cells, further promoting bone formation [241,242]. Additionally, recent studies have demonstrated that BGs have angiogenic properties, promoting the formation of new blood vessels [243]. BGs have also shown antibacterial and anti-inflammatory effects, both in vitro and in vivo, making them valuable for various applications in hard tissue engineering [244,245]. BGs can be used alone or as an inorganic phase in composites or hybrid materials, contributing to their widespread use in the field of bone tissue engineering [246]. The combination of CPCs with BGs provides a promising approach to developing bioactive and functional materials for bone regeneration and repair.

The biological activity of bioactive glasses has been extensively studied, with a focus on the influence of porosity and specific surface area on their performance. When BGs come into contact with physiological fluids, initial ion release occurs, which can lead to a significant increase in pH. This elevated pH level may be detrimental to surrounding cells and tissues [247]. However, the final pH can be controlled by incorporating other ions into the BGs, thereby altering the release rate and concentration of ions in the solution. Trace elements such as strontium (Sr), zinc (Zn), copper (Cu), or cobalt (Co), which are naturally present in the human body, are known for their beneficial effects on bone regeneration [248]. Incorporating these ions into CPCs and BGs can modulate their dissolution behavior and improve the biological performance of the materials [249,250].

To address the limitations of both calcium phosphate cements and BGs and to enhance their in vitro and in vivo properties, there have been attempts to synergistically combine them. The combination of CPCs and BG has been explored in the form of composites, with studies focusing on their physicochemical and osteogenic properties. Bellucci et al. and Karadjian et al. have conducted comprehensive reviews of the literature on CPC and BG composites, highlighting their characteristics and potential applications [251,252]. However, to the best of our knowledge, there has not been a previous systematic review specifically discussing BGs incorporated into injectable CPC bone cement. Further research in this area would be valuable for understanding the synergistic effects and potential applications of these composite materials.

Radiopacity is an important characteristic to consider when developing biomaterials for bone regeneration, as it allows for the visualization of the material during and after surgery. It ensures proper positioning of the biomaterial at the defect site and enables easy detection and monitoring of any potential issues or failures during follow-up. While calcium phosphate cements possess some level of intrinsic radiopacity, they may not be sufficient for precise fluoroscopic control or distinguishing the biomaterial from surrounding bone during surgery [253]. Several systematic studies have been conducted to evaluate the radiopacity of injectable CPCs. These studies have emphasized the importance of investigating the radiopacity of injectable bone cement [254,255,256,257]. In addition to radiopacity, other properties such as mechanical properties, degradation profile, and porosity are important parameters to study in CPC composites [258]. The mechanical properties of CPCs, typically assessed in compression, can be comparable to trabecular bone (4–12 MPa). Some specific CPC formulations have reported compressive strengths of up to 80 MPa for apatite-forming CPC and up to 52 MPa for brushite-forming CPC [259]. However, the inherent fragility of CPCs still limits their clinical use to non-load-bearing applications.

To improve the mechanical properties of CPCs, various approaches have been explored, and one such attempt is modifying the porosity of the cement. It is important to carefully consider the reduction in porosity, as it can influence the biological properties of CPC, including the rate of degradation in vitro and in vivo. The degradation rate should align with the requirements for proper bone regeneration speed. Therefore, it is crucial to achieve a balance among desired material properties, mechanical properties, porosity, and degradation to meet the clinical needs effectively [260,261,262,263,264,265,266,267,268].

The influence of calcium phosphate cements on bone tissue regeneration and growth has been extensively studied, highlighting certain challenges related to the rate of resorption [269,270]. Both slow and rapid rates of resorption can be problematic. A slow resorption rate may hinder osseointegration, while a fast resorption rate can lead to the washing out of CPC fractions from the defect site.

To address the rapid resorption issue, combinations of different calcium phosphates have been proposed [268,269,270,271,272,273]. For instance, the rate of resorption of tricalcium phosphate can be controlled by using two-phase calcium phosphate cements, which help slow down the resorption process. Different compositions of CPCs can elicit different biological reactions based on factors such as CPC chemistry, crystallinity, stoichiometry, dissolution/precipitation behavior, surface chemistry, and porosity. While CPCs demonstrate good osteoconductivity, their effect on osteogenic differentiation is limited due to their relatively low surface reactivity. In vitro studies have used the formation of a surface layer of hydroxyapatite in artificial body fluid (SBF) as an indicator of “biological activity” for materials in contact with bone. However, in vivo studies with DCPD and β-TCP have shown conflicting results regarding their ability to directly bind to bone, despite the formation of an HA layer in SBF.

On the other hand, bioactive glasses have been recognized for their direct binding capability to bone and surrounding tissues. They can serve as an alternative to CPCs. The combination of BGs with calcium phosphate cements has recently emerged, leveraging the binding ability of BGs to bone and tissues. Overall, research in this field continues to explore the optimization of CPC compositions, resorption rates, and the incorporation of bioactive glasses to enhance the biological activity and performance of bone graft substitutes.

The direct connection between calcium phosphate cement and bone occurs through the formation of an HA-like layer on the material’s surface. The initial steps of this process involve a rapid ion exchange, where sodium ions (Na^+^) are exchanged with hydrogen ions (H^+^) and hydronium ions (H_3_O^+^). This is followed by a polycondensation reaction of surface silanols, resulting in the formation of a silica gel layer with a large surface area. Subsequently, nucleation and crystallization of a layer of hydroxycarbonate apatite take place on the surface. This HCA layer closely resembles the mineral phase of bone, enabling the proliferation and differentiation of osteoblasts within this layer to form their extracellular matrix. Thus, incorporating bioactive glasses (BGs) into CPKs is a promising approach to maintaining bioactivity both in vitro and in vivo.

A comparative study by Campion et al [269]. demonstrated that silicate-substituted HA materials exhibited higher biological activity compared to commercially available β- TCP) bone graft substitutes. The study showed that β-TCP exhibited characteristics of octacalcium phosphate with fewer crystals formed on its surface. In contrast, silicate-substituted HA materials had a thick, continuous layer of apatite hydroxycarbonate crystals deposited on their surface. Additionally, one of the main reasons for combining calcium phosphate cements with bioactive glasses is the release of ions from the BGs, which can promote angiogenesis (formation of new blood vessels) and osteogenic differentiation (differentiation of cells into bone-forming cells). This further enhances the biological properties and performance of the composite material [270].

### 2.3. Organic Components of Bone Grafting Materials

In the field of bone engineering and regenerative dentistry, extracellular scaffolds play a crucial role in providing structural support for stem cell attachment and promoting tissue development. These scaffolds are typically made from polymeric materials and serve as matrices that hold regenerative undifferentiated cells while recreating their biological microenvironment. The goal is to facilitate the differentiation, deposition, and mineralization of an extracellular matrix by undifferentiated preosteoblastic and odontoblast cells, eventually replacing the polymeric scaffold structure [273]. 

An ideal tissue scaffold should possess several key characteristics. First and foremost, it should be biocompatible, ensuring that it does not cause adverse reactions or toxicity when in contact with living tissues. Additionally, the scaffold should be biodegradable, meaning it can gradually break down over time and be replaced by newly formed tissue. This controlled degradation is important to match the rate of tissue regeneration. Moreover, the scaffold should be manipulable, allowing for easy shaping and customization to fit specific defect sites or anatomical structures. Tissue scaffolds can be constructed using both synthetic and natural polymers. Synthetic polymers offer advantages such as tunable mechanical properties and controlled degradation rates. However, they often exhibit fewer cellular adhesion sites, lower biological activity, and reduced biocompatibility compared to natural polymers. Natural polymers have gained popularity in tissue engineering due to their availability and similarity to the components of the native extracellular matrix found in connective tissues. Natural polymer-based scaffolds can be derived from proteins such as silk, gelatin, collagen, fibrin, and soy, or polysaccharides such as cellulose, chitosan, and alginate. These natural polymers provide a favorable environment for cell attachment, proliferation, and differentiation, promoting tissue regeneration processes.

Overall, the choice of polymer type for tissue scaffolds depends on specific application requirements, desired properties, and compatibility with the target tissue or organ. Researchers continue to explore and optimize scaffold materials and fabrication techniques to enhance their effectiveness in promoting tissue regeneration and repair [274,275].

#### 2.3.1. Chitosan

Chitin is a widely recognized natural biomaterial with numerous biomedical applications that can be derived from both animal and plant sources. Currently, industrial extraction of chitin primarily involves obtaining it from seashells. Despite its widespread availability and significant functional properties such as biodegradability, bioactivity, and biocompatibility, the poor solubility of chitin limits its utility in tissue engineering. However, the focus can be shifted to chitosan, which is the primary derivative of chitin. Chitosan is a copolymer obtained through the alkaline deacetylation of chitin, composed of d-glucosamine and N-acetyl-d-glucosamine units [276,277]. This natural multifunctional polysaccharide has been extensively researched in the fields of biomedical, pharmaceutical, and tissue engineering. It possesses desirable properties including biodegradability, bioactivity, biocompatibility, and antimicrobial activity [278].

The biomechanical properties of chitosan can be enhanced through copolymerization with synthetic and natural biomaterials (e.g., hydroxyapatite), bioactive osteogenic molecules (e.g., bone morphogenic protein-2 (BMP2)), or polymers (e.g., silk fibrin, collagen, and polycaprolactone). These modifications enable the production of 3D freeze-dried scaffolds, films, and hydrogels, thereby enhancing their practicality for tissue engineering applications [276,279,280,281,282,283]. Furthermore, the addition of bioactive molecules that influence cellular functions and tissue regeneration can further improve the process of bone regeneration [276]. Chitosan microspheres exhibit a spherical structure with diameters ranging from several micrometers to 1000 microns. These microspheres can encapsulate biologically active molecules uniformly within the polymer matrix, enabling a stable and controlled release of these molecules at targeted sites of regeneration. Additionally, novel chitosan-based scaffolds modified with mineral content, BMP, and osteoinductive drugs have been reported to promote stem cell proliferation, adhesion, and differentiation [278,284,285].

In support of this, an animal experiment demonstrated the positive impact of chitosan nanofiber scaffolds on bone regeneration. The implantation of these scaffolds exhibited a beneficial effect by enhancing regenerative bone volume and improving trabecular quality, all without causing any adverse effects. Notably, chitosan nanofibers were found to increase alkaline phosphatase (ALP) activity [282], which is an indicator of osteoblast function, as well as the expression of osteocalcin (OCN) [282]. OCN and ALP are among the biomarkers associated with osteoblasts, playing crucial roles in regulating osteoblast function and facilitating extracellular matrix mineralization during bone remodeling [286,287]. These findings highlight the potential of chitosan nanofibers to promote bone repair by influencing osteoblast activity and facilitating the mineralization process.

#### 2.3.2. Collagen

Collagen, as the primary component responsible for maintaining the structural integrity of tissues, plays a crucial role in bone structure. Type I and V collagen are the main constituents of bone, possessing the ability to bind hydroxyapatite crystals. When properly optimized, types I and II collagen can form collagen fibers that closely mimic the properties of natural extracellular matrix (ECM) collagen [288,289]. Bovine type I collagen sponges have been found to positively affect the proliferation, attachment, and functional activity (such as osteocalcin production) of human osteoblastic cell lines [290].

Hydrogels, which are three-dimensional polymer networks capable of absorbing and retaining water, are highly biocompatible and flexible. Due to their biocompatibility and high water content, hydrogels can serve as effective vehicles for targeted drug delivery in tissue engineering [291]. In a comparative animal study, the implantation of a collagen hydrogel scaffold was shown to promote the regeneration of the periodontal ligament and bone in cases of defects [292]. Collagen scaffolds have demonstrated favorable performance and serve as suitable scaffolds for alveolar bone regeneration, minimizing bone resorption in the alveolar ridge following tooth extraction or sinus augmentation [293,294]. In human trials, collagen scaffolds have been successfully utilized as natural biodegradable carriers for osteoinductive biomaterials and factors, such as human bone morphogenetic protein-2 (BMP2) [295], hydroxyapatite [296], or in combination with bone allografts to preserve alveolar bone and optimize bone regeneration post-tooth extraction [297,298,299]. The incorporation of hydroxyapatite-starch into collagen sponges in a 1:4:10 ratio has been shown to enhance mechanical properties, cell viability, and hematopoiesis.

#### 2.3.3. Hyaluronic Acid

Hyaluronic acid, another natural polymer present in the human body, is predominantly found in connective tissue and serves as a natural glycosaminoglycan within the extracellular matrix, providing a conducive environment for regenerative processes [300]. Numerous studies have demonstrated that hyaluronic acid, when used as a carrier for growth factors, can enhance bone formation [301,302,303]. Sulfated hyaluronic acid, in particular, has been found to suppress osteoclasts and support osteoblasts in diabetic conditions by binding to sclerostin, a potent signaling inhibitor involved in the integration and activity of osteoblasts [304]. These glycoproteins play a pivotal role in stem cell activities such as proliferation and differentiation [305].

Cross-linked hyaluronic acid hydrogels exhibit desirable consistency and mechanical properties for calvarial bone regeneration techniques. The incorporation of tissue particles into hydrogels improves mechanical characteristics, including yield strength and compressive modulus of the graft material. Among these particles, cartilage particles contribute to the highest yield strength, while tendon particles are beneficial for enhancing bone regeneration. In vitro studies have shown that the addition of tendon particles to hyaluronic acid hydrogels promotes superior calcium deposition by osteoblasts [306]. Furthermore, the application of 1% hyaluronic acid gels following tooth extraction accelerates bone repair and regeneration in both healthy and infected sockets [307,308]. A hyaluronic acid-gelatin hydrogel has been successfully utilized as a composite framework plug for one-stage bone grafting into tooth extraction sockets [309]. The combination of chitosan and hyaluronic acid as polyelectrolytes can improve the properties of each polymer, enhance stability, and promote cell adhesion [310]. Chitosan and hyaluronic acid scaffolds have demonstrated the ability to promote osteoblast differentiation by increasing the expression of genes such as collagen αI, osteocalcin, osteopontin, and Runx 2 in preclinical models [300].

#### 2.3.4. Cellulose

Cellulose, the most abundant biopolymer in nature, is commonly found in the cell walls of green plants, but it can also be synthesized by bacteria and fungi. Bacterial cellulose, a nanostructured biopolymer present in bacterial membranes, is composed of nucleotide-activated glucose and holds great potential for applications in tissue engineering, wound healing, and drug delivery [311]. The structure of bacterial cellulose comprises a three-dimensional network of highly oriented nanofibrils, exhibiting remarkable mechanical strength, biodegradability, and antimicrobial properties in its oxidized form [312,313]. Notably, bacterial cellulose exhibits significant similarities to collagen when used in scaffolds. Resorbable bacterial cellulose membranes, treated with electron beam irradiation, have shown comparable efficacy to conventional collagen membranes in regenerating bone defects around implants. These membranes possess similar mechanical properties to collagen membranes but exhibit greater porosity [314].

In animal studies, a 0.1 mm thick bacterial cellulose membrane has been successfully employed to induce bone regeneration in rat calvarium defects, with osteoblasts observed at the periphery and center of the defect [314].

Hydroxypropylmethylcellulose has been utilized as a crosslinking agent in chitosan-based scaffolds. Scanning electron microscopy has revealed significant adhesion of osteoblasts to scaffolds containing 10%, 20%, and 25% hydroxypropyl methylcellulose, indicating that the porous structure of the scaffold provides favorable conditions for cell attachment through cytoplasmic elongation [315].

#### 2.3.5. Soy

Soy protein isolate is a renewable and abundant protein among natural polymers, consisting of over 90% polypeptides that closely resemble the macromolecular structure of natural proteins found in bones [316,317,318]. Salama et al. (2020) conducted a study where they synthesized an oxidized cellulose nanofiber-grafted soy protein hydrolyzate through amidation coupled with an EDC/NHS reaction, specifically for bone tissue engineering purposes [319]. Similarly, in an experiment conducted by Wu et al. (2020), a two-component scaffold consisting of soy protein was used for bone tissue engineering. In vitro cell culture experiments, analyzed using energy-dispersive X-ray spectroscopy, demonstrated that a scaffold composed of 70% soy protein exhibited enhanced cytocompatibility and osteoblastic properties, including improved cell attachment, proliferation, growth, and acceleration of osteogenesis-related gene expression [320].

In animal studies, soy-based biomaterials were compared to synthetic bone grafts (poly(D,L-lactide glycolide) 50:50), and it was found that they induced comparable levels of bone regeneration in the inner, middle, and outer parts of the bone defect [321].

#### 2.3.6. Alginate

Alginate is a biopolymer derived from seaweed, particularly brown algae, and it consists of mannuronic acid and guluronic acid units. One of the notable characteristics of alginate is its ability to be easily modified into various structures such as microspheres, hydrogels, and fibers. This versatility has led to its widespread use in the production of composite scaffolds when combined with other materials like chitosan [322], cellulose [323], and gelatin [324]. Alginate, therefore, serves as an excellent candidate for scaffold materials in tissue engineering applications as well as in drug delivery systems [325].

#### 2.3.7. Silk

Silk proteins have garnered significant attention in bone engineering research due to their excellent mechanical properties. Dignaschi et al. (2016) explored this aspect by incorporating hydroxyapatite minerals into spider silk, resulting in the synthesis of an inorganic-organic hybrid scaffold. In this biomaterial, the spider silk domain contributes to the material’s stability and workability, while the hydroxyapatite binding domain regulates the osteogenic process [326]. A similar study conducted by Hardy et al [327]. in 2016 also investigated Bombyx mori silk modification through the freeze-drying method using decellularized pulp, collagen, and fibronectin. This modification induced significant alkaline phosphatase activity in MG-63 osteoblasts [328]. These studies highlight the potential of silk-based scaffolds in promoting osteogenic properties and bone tissue regeneration.

#### 2.3.8. Carrageenan

Carrageenan is an anionic sulfated polygalactan similar to glycosaminoglycans found in the extracellular matrix, and it occurs naturally in the cell walls of red algae. Its three-dimensional structure promotes the proliferation and adhesion of osteoblasts [329]. When combined with hydroxyapatite, carrageenan exhibits a stimulating effect on osteoblast activity [330]. The addition of carrageenan to the scaffold structure of hydroxyapatite-collagen composite gel enhances its compressive strength [331]. A mixed carrageenan hydrogel with varying ratios of nanohydroxyapatite demonstrates promising performance with minimal cytotoxicity towards human osteoblast cells and significant antimicrobial activity against Pseudomonas aeruginosa [332]. Studies have shown that exposure of cells to nanocomposite carrageenan hydrogel and whitlockite nanoparticles increases the expression of Runt-associated transcription factor-2 proteins and osteopontin [333]. Incorporating carrageenan onto the surface of graphene oxide facilitates the nucleation of hydroxyapatite, while the rough and hydrophilic surface of graphene-carrageenan oxide provides a more favorable structure for cell proliferation [334]. Carrageenan has also been utilized in combination with silk [335], Arabic gum [336], collagen [337], CaCO_3_ particles [338], acrylic acid-graphene [339], gelatin, and chitosan [340] as hybrid bioscaffolds in tissue engineering research focused on bone regeneration. Collagen/nanohydroxyapatite-carrageenan gel has been successfully employed for the delivery of human nerve growth factor beta, promoting inferior alveolar nerve regeneration during distraction osteogenesis at a rate of 0.75 mm/12 h after six days.

#### 2.3.9. Tragacanth Gum

Tragacanth gum is a non-cariogenic polysaccharide derived from plants of the genus Astragalus. It is commonly utilized in tissue engineering to create biodegradable structures, including scaffolds and drug delivery systems [341]. Tragacanth gum scaffolds have shown promising results in bone regeneration applications [342].

While natural polymers offer significant advantages, they often exhibit low stability of mechanical properties, particularly in wet conditions, which is considered a primary drawback for their potential application. To enhance their biomechanical characteristics, natural polymers can undergo modification through cross-linking or complexation procedures with ceramics or metal ions. Green chemicals or natural agents have emerged as a solution for cross-linking natural polymers, improving the mechanical properties of bone scaffolds without compromising their biocompatibility [343,344].

Nanotechnology has also provided a strategy to develop more efficient scaffolds. By reducing the size of structures, surface characteristics can be improved to enhance biocompatibility and tissue growth. Furthermore, beneficial properties like antibacterial activity can be incorporated into these scaffolds. Examples in this field include chitosan-polycaprolactone composites, hydroxyapatite and alginate nanofiber scaffolds, antibacterial chitosan-calcium phosphate composites infused with silver ions, and highly porous chitosan-hyaluronic acid composite scaffolds.

## 3. Inducers of Natural Osteogenesis

A delicate equilibrium between matrix retention and resorption plays a crucial role in regulating the metabolism of the skeletal system. The interplay between osteoblastic and osteoclastic activities is vital for maintaining skeletal function. Age-related bone disorders, such as osteoporosis, typically arise from an upsurge in the resorptive function of osteoclasts [345]. Understanding and regulating the balance between osteoblastic and osteoclastic function is crucial for maintaining skeletal health and preventing age-related bone disorders like osteoporosis. Therapeutic interventions aimed at promoting bone formation and inhibiting excessive bone resorption are important strategies in the management of these conditions. 

### 3.1. Nuclear Factor Kappa-B Ligand Activator Receptor (RANKL)

The nuclear factor kappa-B ligand activator receptor (RANKL) is a ligand produced by various cells, including osteoblasts, that plays a pivotal role in the activation of osteoclasts. Osteoclasts derived from monocytes express receptors for RANKL on their surface. Upon binding of RANKL to these receptors, bone resorption by osteoclasts is initiated. The expression of osteoprotegerin (OPG) by osteoblasts serves to modulate the activity of RANKL and bone resorption. Additionally, inflammatory cytokines, such as those belonging to the interleukin family, are known to enhance osteoclast activity by inducing the expression of RANKL [346]. There is an extensive body of evidence demonstrating the anti-inflammatory, anti-osteoporotic, osteoinductive, and regenerative properties of natural components. Numerous studies have observed that natural food components can impact these processes by inhibiting bone resorption, promoting bone formation and maturation, and consequently enhancing bone regeneration in the presence of bone defects [277,347,348,349].

### 3.2. ”Plant Phenols” and “Polyphenols”

Plant phenols and polyphenols are secondary metabolic compounds that are synthesized through either the shikimate/phenylpropanoid pathway or the malonate/polyketide pathway [350]. Among these compounds, flavonoids play a significant role in bone metabolism and bone formation. They exhibit the ability to promote osteoblastogenesis and can also interfere with osteoclastogenesis, thus potentially preventing bone resorption.

Flavonoids primarily exert their effects by influencing the proliferation and differentiation of mesenchymal stem cells (MSCs), directing them towards osteoblast lineage. They enhance the expression of osteogenic transcription factors and markers through various signaling pathways, including the Wnt and mitogen-activated protein kinase pathways. Activation of these pathways favors osteoblast differentiation from pre-osteoblast cells and MSCs [351].

The mitogen-activated protein kinase pathway is particularly involved in the osteoblast differentiation process mediated by flavonoids. These compounds activate this pathway, leading to the upregulation of osteogenic markers and the promotion of osteoblast differentiation [351]. Overall, flavonoids exhibit a positive influence on bone formation by enhancing osteoblastogenesis and regulating the balance between bone formation and resorption. Their multifaceted effects on various cellular pathways make them promising candidates for the development of therapeutic approaches targeting bone health.

### 3.3. Epigallocatechin-3-Gallate

Camellia sinensis is a plant native to Southeast Asia that has been widely utilized in complementary medicine alongside traditional treatment approaches in various fields of dentistry [352]. The leaf extract of Camellia sinensis contains catechins, making it valuable in dental applications. Among the polyphenols found in C. sinensis, epigallocatechin-3-gallate (EGCG) is the most abundant and has gained popularity in dentistry due to its anti-inflammatory properties [353], antibacterial effects [354], and antioxidant activity [355].

EGCG has been found to effectively stimulate the proliferation, early osteogenic differentiation, and mineralization of primary human dedifferentiated cells [356]. Moreover, it has been shown to enhance the synthesis of osteoprotegerin (OPG) in osteoblasts stimulated by BMP-4 or prostaglandin E2, through the potentiation of p38 mitogen-activated protein kinase and c-Jun stress-activated protein kinase/amino-terminal kinase pathways [357,358]. EGCG also inhibits osteoblast migration induced by insulin and growth factor I, potentially contributing to the regulation of bone remodeling, possibly by suppressing p44/p42 MAP kinase [359]. The ability of EGCG to inhibit osteoblast migration induced by EGF has been proposed as a key mechanism underlying its beneficial effects on proper bone remodeling [360]. Additionally, EGCG has demonstrated the capacity to enhance cell viability in 3D human periosteal cultures and has been used as an osteogenic graft material for periodontal regenerative therapy [360]. Treatment with EGCG at concentrations of 6–10 mM has been shown to upregulate the expression of type I collagen, osteopontin, and osterix in the human periodontal ligament (PDL) cells, suggesting a promising role for this plant polyphenol in periodontal regeneration [361]. However, higher concentrations of EGCG (greater than 10 mM) may have inhibitory effects on the osteogenic differentiation of cells derived from human alveolar bone [362].

Animal experiments have demonstrated that topical administration of EGCG reduces stress-induced premature aging (characterized by the cessation of cell division) in critical-size bone defect cells [363]. Combining human bone morphogenetic protein (BMP) and EGCG as a coating material for biphasic calcium phosphate has shown potential for remodeling and enhancing regeneration in split defects around dental implants [364]. Furthermore, the combination of tricalcium phosphate particles and 0.2 mg EGCG has been shown to stimulate optimal bone regeneration in calvarial defects in animal studies [365]. Other animal and in vitro experiments have indicated that modifying gelatinous and collagenous bone scaffolds and membranes with EGCG can be beneficial for macrophage recruitment [366,367], improving bone-forming ability [368,369], reducing bone resorption [370], and preserving the bone of dental sockets [371].

Overall, the use of EGCG derived from Camellia sinensis holds promise in various aspects of dental applications, including periodontal regeneration, bone remodeling, and enhancement of bone healing processes.

### 3.4. Acemannan

Acemannan is a polydisperse mannan with a b-(1,4)-linked structure that can be derived from the aloe plant [372]. The most cost-effective extraction process of acemannan from aloe vera involves precipitation with cetyltrimethylammonium bromide, as described by Alonso M. et al. [373]. In 2019, Silva S. et al. proposed a modified approach for acemannan using methacrylic anhydride and photocrosslinking under ultraviolet irradiation to enhance its manufacturability, enabling the production of high-value structures [374].

Acemannan has been found to promote the expression of cyclin D1 in cultured fibroblasts through the AKT/mTOR signaling pathway, leading to increased translation of cyclin D1. It also induces the expression of interleukin-6/-8 and p50/DNA binding in gingival fibroblasts via a TLR5/NF-kB-dependent signaling pathway, which plays a crucial role in wound and periodontal healing [375,376,377]. Moreover, acemannan enhances the production of keratinocyte growth factor-1, vascular endothelial growth factor, and type I collagen during the wound-healing process in the oral cavity [378]. In a four-week follow-up study after treating the extraction socket with acemannan gel, significant increases were observed in bone marrow stromal cell proliferation, vascular endothelial growth factor, alkaline phosphatase activity, bone sialoprotein, osteopontin expression, and mineralization [379]. In vivo studies have demonstrated that acemannan-treated groups exhibit higher bone mineral density and faster bone and periodontal ligament regeneration in animals [379,380,381,382].

Furthermore, two randomized controlled trials with 3- and 12-month follow-ups evaluated the effects of acemannan gel treatment on bone regeneration after tooth extraction and apical surgery, respectively. X-ray studies revealed that the administration of acemannan gel significantly improved the rate of bone wound healing without any reported side effects [383,384,385]. These findings highlight the potential of acemannan as a beneficial agent in promoting bone healing and regeneration in dental applications.

### 3.5. Ikariin

Epimedium is the largest genus of herbaceous plants in the Berberidaceae family [386]. Icariin, a prenylated flavonol glycoside, is closely associated with the therapeutic effects of Herba Epimedii. It is known to inhibit the expression of TNF-a, IL-6, and IL-1b in lipopolysaccharide-stimulated inflammatory responses [387,388]. Ikariin has been shown to significantly enhance osteogenic differentiation of MBMS cells. This is evidenced by increased ALP activity and expression of collagen I, osteopontin, and OCN genes, which are regulated through phosphorylation of extracellular signal-regulated kinase, p38 kinase, and N-terminal c-Jun kinase. These three main families of cascades of mitogen-activated protein kinases play a crucial role in osteogenesis [389,390]. Xu X et al. (2019) demonstrated that icariin enhances OCN expression via STAT [391]. Promising results have been obtained by introducing icariin into poly-(ε-caprolactone)/gelatin nanofibers for the synthesis of an artificial composite periosteum. Loading this membrane with icariin at the transplantation site enhances the attachment, proliferation, and differentiation of preosteoblastic cells [392]. Icariin has been utilized in preclinical studies to enhance the osteoinductivity of various biomaterials and tissue scaffolds, such as the submucosa of the small intestine [393], bioactive glass/chitosan [394,395], hydroxyapatite/alginate [361], and poly(lactic-co-glycolic acid)/β-calcium phosphate [396]. In general, these studies have demonstrated improvements in the activation of proteins and genes associated with osteogenesis, the osteogenic activity of preosteoblastic cells, and the regenerative properties of these bone scaffolds. Topical administration of icariin solution has been shown to significantly promote periodontal tissue and alveolar bone regeneration in periodontitis minipigs, as evidenced by computed tomography, histological, and clinical assessments after a 12-week follow-up [396]. Furthermore, the potential enhancement of alveolar bone remodeling with icariin makes it a noteworthy candidate for accelerating tooth movement during orthodontic treatment, although further research is warranted [397].

### 3.6. Curcumin

Curcumin is the primary curcuminoid found in the Curcuma longa plant, which is renowned for its antibacterial, antioxidant, and healing properties [398]. It has been suggested that curcumin may have beneficial effects on various conditions, including modulating inflammatory and oxidative pathways, metabolic syndrome, arthritis, osteoporosis, and other pathologies [399,400]. In a study by Li Y and Zhang ZZ (2018), the osteogenic properties of a collagen/hydroxyapatite scaffold were evaluated under diabetic conditions after incorporating curcumin into the scaffold structure. The results demonstrated that this enhanced scaffold significantly reduced the adverse effects of diabetic serum on the proliferation, migration, and osteogenic differentiation of mesenchymal stem cells (MSCs). Furthermore, topical injection of a curcumin-loaded scaffold into bone defects in diabetic rats resulted in increased bone formation compared to controls [401]. It should be noted that the osteogenic response to curcumin in preosteoblastic cells is concentration-dependent. At a concentration of 1%, increased expression of osteogenic genes and proteins can be observed, while higher concentrations of curcumin may lead to a decrease in cellular activity [402]. Due to its antitumor activity, curcumin can also be used in combination with nanocarriers for bone regeneration in osteosarcoma patients [403,404]. Local release of curcumin from a calcium phosphate matrix can be improved by utilizing poly-(ε-caprolactone) and polyethylene glycol or liposome encapsulation, which enhances its bioavailability and makes it an excellent natural component for bone regeneration around implants [405]. The incorporation of curcumin into an asymmetric collagen membrane has been shown to increase the osteogenic potential at both the transcriptional and translational levels of directed tissue regeneration, imparting antibacterial properties to the modified membrane [406].

### 3.7. Chlorogenic Acid

Chlorogenic acid, an ester of caffeic acid, is present in various herbs, including coffee and beans. Numerous in vitro and in vivo studies have demonstrated the effectiveness of chlorogenic acid as an antibacterial, anti-inflammatory, antitumor, and analgesic agent [407,408,409,410]. In the context of osteogenesis, chlorogenic acid has shown promising results. At a concentration of 30 mM, chlorogenic acid promotes osteogenesis in mesenchymal stem cells derived from human adipose tissue, as evidenced by increased mineralization, alkaline phosphatase activity, and expression of runt-associated transcription factor-2 [411].

Osteoporosis, a prevalent metabolic bone disease, is characterized by decreased bone mass and defects in bone microarchitecture. Imbalances between bone formation and resorption lead to cortical thinning, porosity, and overall bone loss [412]. In bone regenerative processes and biomaterial osseointegration, dysregulation of bone formation and the overexpression of inflammatory and stress response pathways have a significant negative impact [413,414]. Chlorogenic acid has been found to modulate the decline in bone mineral density, increase metabolic markers, and protect bone structure in ovariectomized rats at certain doses. At concentrations of 1 or 10 mM, chlorogenic acid enhances the production of phosphorylated protein kinase B and cyclin D1, as well as the proliferation of bone mesenchymal stem cells in a concentration-dependent manner [415]. Additionally, chlorogenic acid enhances the synthesis of interleukin-6 stimulated by TNF-α in osteoblasts, which plays a crucial role in bone restoration and regeneration following fractures [416,417].

In preclinical settings, the incorporation of 60 mM chlorogenic acid into alginate scaffolds has been shown to stimulate chondrogenesis, chondrocyte proliferation, and cartilage matrix synthesis [418]. These findings highlight the potential of chlorogenic acid as a beneficial herbal ingredient for promoting bone and cartilage regeneration.

### 3.8. Propolis and “Royal Jelly”

Apis mellifera honeybees collect propolis from plants, and this natural product is a complex mixture containing bioactive phenolic acids, flavonoids, and esters. While propolis is an animal product, a significant portion of its functional components originates from plants [419]. Several studies have highlighted the regenerative properties of propolis in bone healing and periodontal treatment. In an in vivo randomized controlled trial with a 56-day follow-up, Meimandi-Parizi et al. demonstrated that transdermal administration of a dilute aqueous propolis extract significantly improved hyaline bone formation and regeneration at defect sites in rats with critical bone defects [420]. Histomorphometric evaluations in dogs have shown that propolis is more effective than nanohydroxyapatite bone graft in regenerating bone in dental furcation defects, with notable improvements in bone height and surface area [421]. Propolis has also exhibited a protective effect against osteopathy in diabetic conditions [422].

The combination of propolis and bovine bone graft has been found to increase the expression of heat shock protein and osteocalcin, as well as the number of osteoblast cells in sockets after tooth extraction compared to bovine graft alone [423]. Systemic administration of propolis in albino rats stimulated osteoblast concentration and bone formation in premaxillary sutures undergoing orthopedic expansion [424].

Royal jelly, another natural product produced by Apis mellifera honeybees, is rich in proteins, royalactin, lipids, and vitamins such as pantothenic acid [425]. In an in vitro study by Yanagita et al. (2011), treatment of mouse periodontal ligament cells with raw royal jelly resulted in increased expression of osteopontin, osteocalcin, and osterix mRNA, as well as enhanced mineralized nodule formation. These findings suggest that royal jelly holds promise as a suitable candidate for regenerative periodontal treatment protocols [426]. Caffeic acid phenethyl ester, an active ingredient in propolis, has also been investigated for its healing properties. Intraperitoneal injection of this bioactive compound into animal bone defects and sockets after tooth extraction has been shown to accelerate healing [427,428].

Overall, propolis, royal jelly, and caffeic acid phenethyl ester offer potential therapeutic benefits for bone regeneration and periodontal treatment, making them valuable natural products for further exploration in the field of regenerative medicine.

### 3.9. Salvia miltiorrhiza

*Salvia miltiorrhiza*, a perennial plant, possesses a wide range of medicinal properties. The plant contains hydrophilic phenols such as salvianolic acids, lipophilic diterpenoids, flavonoids, and triterpenoids, which are its main bioactive compounds [429]. The treatment of osteoblast-like cell clones with *S. miltiorrhiza* induces the rapid expression of alkaline phosphatase, indicating enhanced bone remodeling [430]. This plant regulates the expression of alkaline phosphatase, osteocalcin, osteoprotegerin, and receptor activator of nuclear factor-kappa B ligand (RANKL) genes, further suggesting its role in bone remodeling. Salvianolic acid, derived from *S. miltiorrhiza*, stimulates the differentiation of bone marrow stromal cells into osteoblasts and activates their functions [431]. This phenolic compound promotes the osteogenic activity of mesenchymal stem cells (MSCs) through kinase signaling pathways regulated by extracellular signals, without exhibiting cytotoxicity [432]. In human periodontal ligament cells, salvianolic acid induces osteogenic differentiation via the Wnt/β-catenin signaling pathway [433].

The combination of *S. miltiorrhiza* extract and MSCs has been reported to promote the revascularization of avascular necrotic bone by upregulating the expression of bone morphogenetic protein 2 (BMP2) and vascular endothelial growth factor, leading to reossification and revascularization [434]. When used in combination with a collagen matrix in rat calvarial defect models, *S. miltiorrhiza* extract increases bone formation activity. However, histological analysis revealed the presence of multinucleated giant cells, indicating a foreign body reaction in the defect area [435].

Tanshinol, an aqueous polyphenol isolated from *S. miltiorrhiza* Bunge, exhibits inhibitory effects on osteoclastogenesis and counteracts glucocorticoid-induced osteoporosis and oxidative stress. It achieves this by inhibiting bone marrow adiposity through the KLF15/PPARγ2/FoxO3a/Wnt/NF-κB pathways [436,437,438,439,440].

In summary, *Salvia miltiorrhiza* and its bioactive compounds, including salvianolic acid and tanshinol, have demonstrated positive effects on bone health and regeneration. These findings highlight the potential of *S. miltiorrhiza* as a valuable natural resource for the development of therapies targeting bone-related disorders.

### 3.10. Resveratrol

Resveratrol, a derivative of polyphenol and natural stilbene found in food resources such as berries, grapes, nuts, and cocoa, has been studied for its potential benefits in bone tissue engineering and bone health [441]. The formation of mature vasculature within the scaffold is crucial in bone tissue engineering, and resveratrol has shown promising effects in this regard. In an animal study, it was reported that resveratrol can prevent steroid-induced osteonecrosis by improving the blood supply to the bone structure [442]. Resveratrol induces the expression of vascular endothelial growth factor, mannose receptor C-type 1, bone morphogenetic protein 2, and alkaline phosphatase activity in human mesenchymal stem cells, irrespective of inflammation [443,444,445]. It also promotes osteogenic differentiation of both embryonic and induced pluripotent stem cells, protects stem cell-derived osteocyte-like cells from glucocorticoid-induced oxidative damage, and reduces the tumorigenicity of pluripotent stem cells [445].

Pretreatment of MSCs derived from human adipose tissue with resveratrol prior to seeding into 3D tissue-engineered structures has been shown to induce the production of a mineralized matrix [446]. Resveratrol exhibits antioxidant action and is able to inhibit alveolar bone and periodontal destruction in rat models of periodontitis [447,448]. In an animal study, intraperitoneal injection of resveratrol at a concentration of 10 mmol/kg significantly improved bone regeneration after tooth extraction [449]. Preclinical studies have evaluated the regenerative capacity of various bone and cartilage tissue scaffolds, such as collagen, chitosan, poly-ε-caprolactone, poly-caprolactone, and hyaluronic acid, after enrichment with resveratrol [450,451,452,453,454,455]. Most of these studies have shown promising results in terms of enhancing tissue regeneration.

It is important to note that while resveratrol may reduce bone loss in older animals after oophorectomy, it does not have a protective effect and may have a negative effect on bone health at an early age [456,457]. Further research is needed to understand the age-dependent effects of resveratrol on bone health.

In summary, resveratrol exhibits potential benefits in bone tissue engineering, including promoting vasculature formation, enhancing osteogenic differentiation, and improving bone regeneration. However, its effects may vary depending on age and other factors, and more research is needed to fully understand its impact on bone health.

### 3.11. Rutin

*Morinda citrifolia*, also known as noni, is a traditionally used plant with healing properties for bone fractures and connective tissue regeneration enhancement, as well as immunomodulation [458,459]. These beneficial effects have been attributed to the presence of rutin, a bioflavonoid commonly found in this plant species [460]. In vitro studies have demonstrated that *M. citrifolia* fruit juice promotes the proliferation of bone marrow stem cells and induces marker genes associated with osteogenic differentiation [458]. Furthermore, the extract derived from the plant’s leaves can stimulate osteogenic activity and mineralization in human periodontal cells and gingival stem cells through the activation of the PI3K/Akt-dependent Wnt/b-catenin signaling pathway, without causing cytotoxicity [461,462].

Rutin, the bioactive compound found in *M. citrifolia*, has been reported to retain its osteogenic properties even in inflammatory conditions by inhibiting the release of reactive oxygen species [463]. In animal studies using estrogen-deficient rats, *M. citrifolia* aqueous extract has shown concentration-dependent improvements in bone density, structure, flexibility, and strength, with the optimal effects observed at a concentration of 300 mg/kg body weight. These effects are attributed to the enhancement of osteoblast activity and the inhibition of osteoclast activity [464]. Moreover, preclinical studies have demonstrated that rutin exerts osteogenic and protective effects on periodontal ligament (PDL) cells through the activation of the PI3K/AKT/mTOR pathway. It also counteracts TNF-alpha-induced damage to the osteogenic differentiation of these cells [465,466]. Combining rutin with vitamin C has been suggested to enhance its osteogenic properties [467], as both compounds have been independently associated with bone health and regeneration.

In summary, *Morinda citrifolia* and its bioactive compound rutin have shown promising osteogenic properties and have the potential to enhance bone regeneration and protect against bone-related conditions. Further research is needed to explore their mechanisms of action and evaluate their efficacy in clinical settings.

### 3.12. Osthole

Osthole, a bioactive coumarin derivative found naturally in plants like Cnidium monnieri, has been studied for its osteogenic properties and its potential in bone healing processes. In mouse preosteoblastic cells, Osthole has been shown to induce osteogenesis by modulating the cAMP/cAMP response element signaling pathway. It promotes the downstream expression of the transcription factor Sterix and inhibits the activity of RANKL, which is involved in osteoclast-mediated bone resorption [468,469].

Furthermore, Osthole has been found to accelerate the process of endochondral ossification and bone fracture healing. It does so by inducing the expression of cartilage marker genes and bone morphogenetic protein 2 (BMP2), a key regulator of bone formation [470,471]. In periodontal ligament-derived stem cells, Osthole has shown the ability to activate the acetylation of Histone 3 lysine 9 and Histone 3 lysine 14. These modifications play a crucial role in the osteogenic differentiation of periodontal cells, suggesting that Osthole may enhance bone regeneration in the context of periodontal tissue [472,473].

Overall, Osthole exhibits osteogenic properties by regulating signaling pathways, promoting cartilage and bone marker expression, and influencing the differentiation of stem cells involved in bone formation. These findings highlight the potential of Osthole as a natural compound for promoting bone health and regeneration. However, further research is needed to fully understand its mechanisms of action and evaluate its efficacy in clinical settings.

## 4. Conclusions

Bone, despite its relatively simple structure composed of three types of cell subpopulations, an organic part mainly consisting of collagen, and an inorganic phase primarily composed of hydroxyapatite, presents a challenge in the development of an “ideal” bone grafting material due to the complex regulation of bone metabolism and the composite nature of its extracellular matrix. However, the application of various calcium–phosphate materials, particularly hydroxyapatite and polymer/hydroxyapatite composites, has shown relatively successful outcomes in experimental and clinical settings. It is important to note that the properties and clinical efficacy of these materials are strongly influenced by synthesis features such as the Ca/P ratio, the presence of amorphous calcium phosphates, and structural parameters like grain size and porosity. In addition to collagen, which serves as the main component of native bone tissue, other materials such as chitosan, gelatin, and alginates, among others, have demonstrated the ability to promote bone regeneration and provide additional benefits as antibacterial and vascular growth-controlling factors. Moreover, natural compounds have been explored as additives to bone grafting materials, aiming to regulate bone healing processes. Recent advancements in bone grafting material discovery have showcased their potential to offer control over bone regeneration, reduce the risk of disease transmission, and expedite the recovery process following application. Looking ahead, future research should focus on standardizing the production of bone grafting materials, controlling parameters such as porosity and mechanical properties, and tailoring their characteristics to enable precise programming of degradation and facilitate new bone growth.

## Figures and Tables

**Figure 1 polymers-15-03822-f001:**
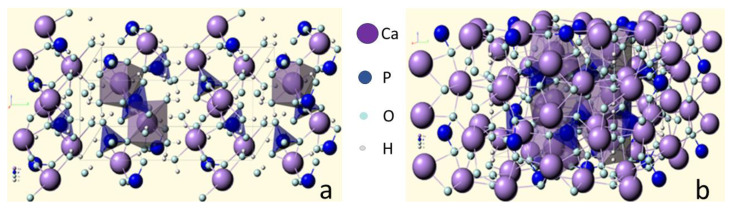
Crystal structure of (**a**) brushite and (**b**) hydroxyapatite [48,68,69].

**Table 1 polymers-15-03822-t001:** Existing calcium phosphates and their main properties [55,56,57].

Ca/P Atomic Ratio	Compound	Formula	Solubility at 25 °C, −log(Ks)	Solubility at 25 °C, g/L	Stability pH in Water Solutions at 25 °C
0.5	Calcium dihydrophosphate monohydrate (MCPM)	Ca(H_2_PO_4_)_2_·H_2_O	1.14	~18	0.0–2.0
0.5	Calcium dihydrophosphate anhydrous (MCPA)	Ca(H_2_PO_4_)_2_	1.14	~17	Stable at >100 °C
1.0	Calcium hydrophosphate anhydrous (DCPA), mineral monetite	CaHPO_4_	6.90	~0.048	Stable at >100 °C
1.0	Calcium hydrophosphate dihydrate (DCPD), mineral brushite	CaHPO_4_ ·2H_2_O	6.59	~0.088	2.0–6.0
1.33	Octacalcium phosphate (OCP)	Ca_8_(HPO_4_)_2_(PO_4_)_4_·5H_2_O	96.6	~0.0081	5.5–7.0
1.5	α-tricalcium phosphate (α-TCP)	α-Ca_3_(PO_4_)_2_	25.5	~0.0025	Obtained at solid state
1.5	β-tricalcium phosphate (β-TCP)	β-Ca_3_(PO_4_)_2_	28.9	~0.0005	Obtained at solid state
1.0–2.2	Amorphous calcium phosphate (ACP)	Ca_x_H_y_(PO_4_)_z_·nH_2_O n = 3–4.5; 15–20% H_2_O	*	*	~5–12 (always metastable)
1.5–1.67	Calcium deficient hydroxyapatite (CDHA) (as prepared HA)	Ca_10−_*_x_*(HPO_4_)*_x_*(PO_4_)_6−_*_x_*(OH)_2−_*_x_* At x = 1 (0 < *x* < 1) Ca_9_(HPO_4_)(PO_4_)_5_(OH)	~85.1	~0.0094	6.5–9.5
1.67	Hydroxyapatite (HA, HAp or OHAp)	Ca_10_(PO_4_)_6_(OH)_2_	116.8	~0.0003	9.5–12
1.67	Fluorapatite (FA or FAp)	Ca_10_(PO_4_)_6_F_2_	120.0	~0.0002	7–12
1.67	Oxyapatite (OA or OAp)	Ca_10_(PO_4_)_6_O	~69	~0.087	Obtained at solid state
2.0	Tetracalcium phosphate (TTCP), mineral hilgenstokite	Ca_4_(PO_4_)_2_O	38–44	~0.0007	Obtained at solid state

* The exact measurement is not possible, but the following data were found: 25.7 ± 0.1 (pH = 7.40), 29.9 ± 0.1 (pH = 6.00), and 32.7 ± 0.1 (pH = 5.28). Comparison of solubility in acetate buffer ACP >> α-TCP >> β-TCP > CDHA >> HA > FA.

**Table 2 polymers-15-03822-t002:** The ways of hydroxyapatite synthesis from aqueous solutions [48,81].

Initial Reactants	Parameters of Synthesis	Morphology
Precipitation		
Ca(NO_3_)_2_ (NH_4_)_2_HPO_4_ (0.5 M, 1 L)	pH = 9.5 (NH_4_OH), 25 °C, 24 g, Ca/P = 1.5	Agglomerate (10–80 µm), from granules (0.06 µm)
Ca(NO_3_)_2_ (1 M) (NH_4_)_2_HPO_4_ (1 M)	pH = 7–11 (NH_3_, NH_4_NO_3_), Ca/P = 1.5–1.67	Surface area 116–119 m^2^/g
Ca(NO_3_)_2_ (0.13 M, 2.5 L) (NH_4_)_2_HPO_4_ (0.07 M, 2.5 L) CH_3_COONH_4_ (1 N)	pH = 8.5–9.5 (NH_3_), 100 °C, >5 g, Ca/P = 1.68	Granules (5 µm)
Ca(NO_3_)_2_ (0.13 M, 2.5 L) (NH_4_)_2_HPO_4_ (0.07 M, 2.5 L) CH_3_COONH_4_ (1 N)	pH = 3.5–9.5 (NH_3_), 100 °C, >5 g, Ca/P = 1.73	Whiskers (1.9 × 0.14 µm)
Na_2_HPO_4_ (0.3 M) CaCl_2_ (0.5 M)	pH = 8.5–9.5 (NH_4_OH), 70 °C, 24 g, Ca/P = 1.65	Surface area 39.7 m^2^/g
H_3_PO_4_ (0.5 M, 4 L) CaCl_2_ (0.5 M, 7 L)	100 °C, 18 g, Ca/P = 1.67	Surface area 16.7 m^2^/g
NaH_2_PO_4_ (0.1 M, 2 L) Ca(NO_3_)_2_ (0.167 M, 2 L)	pH = 8.5 (NaOH), 95 °C, 24 h, Ca/P = 1.67	Granules (0.025 µm)
Ca(OH)_2_ H_3_PO_4_ (0.5 M)	pH = 7.95 °C, 2–6 days, Ca/P = 1.67	40–60 × 60–90 nm
Ca(NO_3_)_2_ (0.01–1 M) (NH_4_)_2_HPO_4_ (0.01–1 M)	pH = 7–10 (NH_3_), Ca/P = 1.5–1.67	Flat needles 15–30 × 25–70 nm
CaHPO_4_·2H_2_O (120 g) H_2_O (4 L)	pH = 8.5 (NH_3_), 100 °C, >0.5 g, Ca/P = 1.67	Granules (75 µm)
CaHPO_4_·2H_2_O (80 g) H_2_O (0.4 L)	pH = 8.5 (NH_4_OH), 40 °C, 3 r, Ca/P = 1.51	Plates with whiskers (0,1 µm)
α-Ca_3_(PO_4_)_2_ (40 g) H_2_O (1 L)	pH = 5.5–10 (NH_4_OH), (HNO_3_) 80 °C, 2–3 g, Ca/P = 1.5–1.68	Agglomerates (10–30 µm) flakes (2 × 2 µm) and needles (5 × 0.2 µm)
α-Ca_3_(PO_4_)_2_ + Ca(OH)_2_, HNO_3_	pH = 4–10, 95 °C, 2–3 g, Ca/P = 1.67	Plates, needles (3–5 µm)

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
