# Peer review of "Synthetic Calcium–Phosphate Materials for Bone Grafting"

_polymers, 2023, doi:10.3390/polym15183822_

Round 1

Reviewer 1 Report

The authors have written and organized the content of the review article very well. I have few suggestions for the manuscript before publication.

Figure 1, font of subfigure “b” needs correction.

Move heading “Heat treatment of precursors and preparation of final fine ceramic CaP powders” from page 12 to page 13

The section 2.1 (from 2.1.1 to 2.1.7) is discussion regarding different bone grafting materials, while rest 2.2., 2.3., etc are guiding principles.

After 2.1.1. Brushite, there is no heading till 2.1.7, the discussion in subsection directly starts with bold first letter, kindly define the bold headings properly.

Section 2.5, the bold text should be made normal or heading properly to avoid confusion. E.g., page 17, “In vivo degradation”; Page 18, “The presence of pores”; Page 109, “Porosity and interconnectivity” etc…

On page 22, few lines are made bold, author can make it normal or highlight appropriately by making proper heading if required. Same trend can be seen throughout manuscript and needs modification.

For the sake of simplicity, the structure of review can be considered to modify as

1. Introduction

2. General principles of bone substitute synthesis

2.1. Inorganic phases of bone substitute materials.

2.1.1. Inorganic Materials:

a. Calcium Phosphate

b. Brushite

2.1.2. Guiding Principles:

a. XX

b. YY

2.2. Organic components of bone grafting materials: Extracellular scaffolds

a. Chitosan

b. Collagen

3. Inducers of natural osteogenesis

4. Conclusions

Author Response

Manuscript ‘Synthetic Calcium-phosphate Materials for Bone Grafting’     paper by Oleg Mishchenko, Anna Yanovska, Oleksii Kosinov, Denys Maksymov, Roman Moskalenko, Arunas Ramanavicius, Maksym Pogorielov

After recommended revisions

Response to reviewer #1:

We would like to thank the reviewer for the review of our manuscript, valuable comments and recommendations. We did our best in order to improve the manuscript according to minor revisions recommended by all three reviewers. All the most important changes are highlighted in the revised manuscript.

Reviewer #1 wrote: The authors have written and organized the content of the review article very well. I have few suggestions for the manuscript before publication.

Response to the reviewer: We will thank the reviewer for positive comments on Significance Level and recommendations.

Reviewer #1 wrote: Figure 1, font of subfigure “b” needs correction.

Response to the reviewer: Dear reviewer, according to your recommendation, the Figure 1 was completely corrected also with subfigure fonts.

Reviewer #1 wrote: Move heading “Heat treatment of precursors and preparation of final fine ceramic CaP powders” from page 12 to page 13.

Response to the reviewer: Dear reviewer, we moved the heading to page 13.

Reviewer #1 wrote: 3. The section 2.1 (from 2.1.1 to 2.1.7) is discussion regarding different bone grafting materials, while rest 2.2., 2.3., etc are guiding principles.

Response to the reviewer: Dear reviewer, it was corrected according your recommendation.

Reviewer #1 wrote: After 2.1.1. Brushite, there is no heading till 2.1.7, the discussion in subsection directly starts with bold first letter, kindly define the bold headings properly.

Response to the reviewer: Dear reviewer, following subsections were added:

2.1.1. Brushite

2.1.2. Calcium-deficient apatite (CDA)

2.1.3. Octacalcium phosphate (OCP)

2.1.4. Amorphous calcium phosphate (ACP)

2.1.5. Tricalcium phosphate (TCP)

2.1.6. Stoichiometric HA

2.1.7. Fluorapatite (FA)

Reviewer #1 wrote: Section 2.5, the bold text should be made normal or heading properly to avoid confusion. E.g., page 17, “In vivo degradation”; Page 18, “The presence of pores”; Page 109, “Porosity and interconnectivity” etc…

Response to the reviewer: Dear reviewer, we removed the bold text as your recommended.

Reviewer #1 wrote: On page 22, few lines are made bold, author can make it normal or highlight appropriately by making proper heading if required. Same trend can be seen throughout manuscript and needs modification.

Response to the reviewer: Dear reviewer, the bold text was removed.

Reviewer #1 wrote: For the sake of simplicity, the structure of review can be considered to modify as

  1. Introduction
  2. General principles of bone substitute synthesis

2.1. Inorganic phases of bone substitute materials.

2.1.1. Inorganic Materials:

  1. Calcium Phosphate
  2. Brushite

2.1.2. Guiding Principles:

  1. XX
  2. YY

2.2. Organic components of bone grafting materials: Extracellular scaffolds

  1. Chitosan
  2. Collagen
  3. Inducers of natural osteogenesis
  4. Conclusions

Response to the reviewer: Dear reviewer, the structure was modified as follows:

  1. Introduction
  2. General principles of bone substitute synthesis

2.1. Inorganic phases of bone substitute materials. Calcium phosphate materials

2.1.1. Brushite

2.1.2. Calcium-deficient apatite (CDA)

2.1.3. Octacalcium phosphate (OCP)

2.1.4. Amorphous calcium phosphate (ACP)

2.1.5. Tricalcium phosphate (TCP)

2.1.6. Stoichiometric HA

2.1.7. Fluorapatite (FA)

2.2. Guiding Principles:

2.2.1. Heat treatment of precursors and preparation of final fine ceramic CaP powders.

2.2.2. Ordered Mesoporous Silicon-Calcium-Phosphate Composites

2.2.3. Growth of hydroxyapatite nanoparticles in ordered mesoporous silica.

2.2.4. Requirements for calcium phosphate cements (CPCs)

2.2.5. Foam concentrates for increasing porosity of calcium phosphate cements

2.2.6. Calcium phosphate (CaP) ceramic based bone grafts.

2.3. Organic components of bone grafting materials

2.3.1. Chitosan

2.3.2. Collagen

2.3.3. Hyaluronic acid

2.3.4. Cellulose

2.3.5. Soy

2.3.6. Alginate

2.3.7. Silk

2.3.8.  Carrageenan

2.3.9. Tragacanth gum

  1. Inducers of natural osteogenesis

3.1.- …..3.12

  1. Conclusions

Many thanks for the comment, the related improvement was performed.

Many thanks for the positive feedback.

We thank you for the attention you will pay to this revised version of the manuscript and we sincerely hope that our work after these revisions will be considered as relevant and attractive for publishing.

Yours sincerely,

Arunas Ramanavicius

----------------------------------------------------------------
Prof. habil. dr. Arunas Ramanavicius

Head of Department of Physical Chemistry,

Faculty of Chemistry and Geosciences, Vilnius University, 

e-mail: arunas.ramanavicius@chf.vu.lt

Reviewer 2 Report

The title of manuscript is good. English language has good quality. Tables need some changes. There are some explainations about different parts of the manuscript that rhe authors should answer to them.

1. In page 5, Table 1

Please insert a column about the biological or grafting usages of each type of alcium phosphates

2. In page 10, Table 2

Please determine the source of aqueous solutions for each column

3. Please write about animal studies of usages of Calcium-phosphate Materials for tissue grafting

4. Please write about future insights and

obstackles of your present work

5. Please determine the details of Figure 1

6 . What about in-silico studies that are related to your present work? Is there any in-silico studies that has tried to clarify the role of Calcium-phosphate Materials in tissue grafting?

7. The title of present manuscript is "Synthetic Calcium-phosphate Materials for Bone Grafting" therefore, please explain why you have written a section entitled " 3. Extracellular scaffolds – organic components of bone grafting materials" in page 25?

8. Please insert a simple, schematic figure about the gist of your manuscript

9. Please write a separate section about

serious liabilities of Synthetic Calcium-phosphate Materials in tissue grafting

10. Please write a discrete section about the applications of Synthetic Calcium-phosphate Materials in other types of tissue Grafting

11. Please check and adjust the "Reference list" based on the regulations of reference list of journal. (Titles, doi, the name of journal and ... )

Author Response

Manuscript ‘Synthetic Calcium-phosphate Materials for Bone Grafting’     paper by Oleg Mishchenko, Anna Yanovska, Oleksii Kosinov, Denys Maksymov, Roman Moskalenko, Arunas Ramanavicius, Maksym Pogorielov

After recommended revisions

Response to reviewer #2:

We would like to thank the reviewer for the review of our manuscript, valuable comments and recommendations. We did our best in order to improve the manuscript according to minor revisions recommended by all three reviewers. All the most important changes are highlighted in the revised manuscript.

Reviewer #2 wrote: The title of manuscript is good. English language has good quality. Tables need some changes. There are some explanations about different parts of the manuscript that the authors should answer to them.

Response to the reviewer: We will thank the reviewer for positive comments on Significance Level and recommendations.

Reviewer #2 wrote: In page 5, Table 1 Please insert a column about the biological or grafting usages of each type of calcium phosphates

Response to the reviewer: Dear reviewer, due to the lack of place in the table, this information were added in the text below the table 1. It is pointed by yellow P.6-10.

Reviewer #2 wrote: In page 10, Table 2 Please determine the source of aqueous solutions for each column

Response to the reviewer: Dear reviewer, all sources are given in details for this table in references 48 and 82.

Reviewer #2 wrote: Please write about animal studies of usages of Calcium-phosphate Materials for tissue grafting.

Response to the reviewer: Dear reviewer, P.3. P.16, P.18-19 required information was added.

Reviewer #2 wrote: Please write about future insights and vobstackles of your present work.

Response to the reviewer: Dear reviewer, thank you for this valuable coment. We provide short insight in conclusion section.

Reviewer #2 wrote: Please determine the details of Figure 1.

Response to the reviewer: Dear reviewer, figure was completely corrected, details were added

Reviewer #2 wrote: What about in-silico studies that are related to your present work? Is there any in-silico studies that has tried to clarify the role of Calcium-phosphate Materials in tissue grafting?

Response to the reviewer: Dear reviewer, in this work we have no in-silico studies. It is not connected with our present work.

Reviewer #2 wrote: The title of present manuscript is "Synthetic Calcium-phosphate Materials for Bone Grafting" therefore, please explain why you have written a section entitled " 3. Extracellular scaffolds – organic components of bone grafting materials" in page 25?

Response to the reviewer: Dear reviewer, we changed section 2.3. into Organic components of bone grafting materials. We propose to change the name of the article into  "Synthetic and Natural Materials for Bone Grafting"

Reviewer #2 wrote: Please insert a simple, schematic figure about the gist of your manuscript

Response to the reviewer: Dear reviewer, we kindly thank you for this recommendation. Taking into account that this is a review paper we did not prepare the graphical abstract. We hope that the absence of the graphical abstract will not affect the quality of the final version. 

Reviewer #2 wrote: Please write a separate section about serious liabilities of Synthetic Calcium-phosphate Materials in tissue grafting. Please write a discrete section about the applications of Synthetic Calcium-phosphate Materials in other types of tissue Grafting

Response to the reviewer: Dear reviewer, thank you for this suggestion. The Synthetic Calcium-phosphate Materials is widely used in bone tissue grafting as well as in preparation of some other graft for soft tissues. Unfortunately, our manuscript is already about 70 pages long with 475 references and adding more section will significantly increase their length. This topic is significant for research society and in the near future we are going to prepare the separate material about the application of Synthetic Calcium-phosphate Materials in tissue grafting.

Reviewer #2 wrote: Please check and adjust the "Reference list" based on the regulations of reference list of journal. (Titles, doi, the name of journal and ... )

Response to the reviewer: Dear reviewer, reference list was corrected.

Many thanks for the comment, the related improvement was performed.

Many thanks for the positive feedback.

We thank you for the attention you will pay to this revised version of the manuscript and we sincerely hope that our work after these revisions will be considered as relevant and attractive for publishing.

Yours sincerely,

Arunas Ramanavicius

----------------------------------------------------------------
Prof. habil. dr. Arunas Ramanavicius

Head of Department of Physical Chemistry,

Faculty of Chemistry and Geosciences, Vilnius University, 

e-mail: arunas.ramanavicius@chf.vu.lt

Round 2

Reviewer 1 Report

The manuscript looks good now and is recommended for publication.

Reviewer 2 Report

No more suggestion.